# Overall photosynthesis of H$_2$O$_2$ by an inorganic semiconductor

Tian Liu[1], Zhenhua Pan [2✉], Junie Jhon M. Vequizo [3], Kosaku Kato [4], Binbin Wu [1], Akira Yamakata [4], Kenji Katayama[2], Baoliang Chen [1], Chiheng Chu [1✉] & Kazunari Domen [3,5]

Artificial photosynthesis of H$_2$O$_2$ using earth-abundant water and oxygen is a promising approach to achieve scalable and cost-effective solar fuel production. Recent studies on this topic have made significant progress, yet are mainly focused on using organic polymers. This set of photocatalysts is susceptible to potent oxidants (e.g. hydroxyl radical) that are inevitably formed during H$_2$O$_2$ generation. Here, we report an inorganic Mo-doped faceted BiVO$_4$ (Mo:BiVO$_4$) system that is resistant to radical oxidation and exhibits a high overall H$_2$O$_2$ photosynthesis efficiency among inorganic photocatalysts, with an apparent quantum yield of 1.2% and a solar-to-chemical conversion efficiency of 0.29% at full spectrum, as well as an apparent quantum yield of 5.8% at 420 nm. The surface-reaction kinetics and selectivity of Mo:BiVO$_4$ were tuned by precisely loading CoO$_x$ and Pd on {110} and {010} facets, respectively. Time-resolved spectroscopic investigations of photocarriers suggest that depositing select cocatalysts on distinct facet tailored the interfacial energetics between {110} and {010} facets and enhanced charge separation in Mo:BiVO$_4$, therefore overcoming a key challenge in developing efficient inorganic photocatalysts. The promising H$_2$O$_2$ generation efficiency achieved by delicate design of catalyst spatial and electronic structures sheds light on applying robust inorganic particulate photocatalysts to artificial photosynthesis of H$_2$O$_2$.

[1] Faculty of Agriculture, Life, and Environmental Sciences, Zhejiang University, 310058 Hangzhou, China. [2] Department of Applied Chemistry, Faculty of Science and Technology, Chuo University, 1-13-27 Kasuga, Bunkyo, Tokyo 112-8551, Japan. [3] Research Initiative for Supra-Materials, Shinshu University, 4-17-1 Wakasato, Nagano-shi, Nagano 380-8553, Japan. [4] Graduate School of Engineering, Toyota Technological Institute, 2-12-1, Hisakata, Tempaku, Nagoya 468-8511, Japan. [5] Office of University Professors, The University of Tokyo, 2-11-16 Yayoi, Bunkyo, Tokyo 113-8656, Japan. ✉email: zhenhua.20y@g.chuo-u.ac.jp; chuchiheng@zju.edu.cn

Harvesting solar fuels by artificial photosynthesis has great values in the global missions on tackling climate change and environmental pollutions[1–3]. Among various artificial photosynthetic reactions, solar-driven water splitting for hydrogen generation has attracted the most attention in the past half century. Yet, its practical application is challenged by the low-energy density, storability, and transportability of hydrogen gas[4,5]. To this end, photosynthesis of $H_2O_2$, an emerging liquid fuel and also a green oxidant, is attracting growing interests[6]. Among primary photosynthetic systems, including photovoltaic-assisted electrolysis[7], photoelectrochemical catalysis[8,9], and particulate photocatalysis (PC)[10], PC is the most cost-effective because of its simplicity and scalability[11]. With regard to reaction processes, PC systems are advantageous for the mass transport of reagents and products, which greatly reduces the concentration overpotential and pH gradient during reactions[12]. For these reasons, it is desirable to develop efficient PC systems for $H_2O_2$ generation.

Recently, various PC systems based on organic-polymer semiconductors have been developed for photocatalytic $H_2O_2$ generation with a recording solar-to-$H_2O_2$ (STH) conversion efficiency of 0.61%[13–15]. Nevertheless, these semiconductors have a potential concern of their stability since photocatalytic $H_2O_2$ generation is inevitably accompanied by hydroxyl radical (•OH) generation ($H_2O_2 + h\nu \rightarrow 2$•OH or $H_2O_2 + e^- + H^+ \rightarrow$ •OH + $H_2O$) and such a potent oxidant ($E^0$ (•OH/$H_2O$) = 2.18 V vs. NHE at pH 7.0) is damaging to organic structures[16]. For instance, after 24-h incubation under •OH-rich conditions, graphitic carbon nitride ($C_3N_4$, one of the most widely studied organic photocatalysts for $H_2O_2$ photosynthesis) lost over 60% activity for $H_2O_2$ generation (Fig. S1). In contrast, inorganic semiconductors (e.g., BiVO₄) are resistant to •OH-mediated oxidation, so they are more favored by long-term reactions. Yet, inorganic semiconductors remain inefficient for photocatalytic $H_2O_2$ generation (<150 µM/h, see Table S1) due to high-charge recombination[13,17]. For an efficient inorganic photocatalyst, it needs to exhibit (i) a suitable band structure for $O_2$ reduction and $H_2O$ oxidation coupled with a narrow band-gap, (ii) efficient charge separation, and (iii) high surface-reaction kinetics and selectivity.

Here, we report a faceted Mo-doped BiVO₄ (Mo:BiVO₄) with dual cocatalysts selectively loaded on its reduction and oxidation facets (Fig. 1a). BiVO₄ is a photocatalyst with a suitable band structure and relatively narrow band-gap (2.4 eV) for $H_2O_2$ photosynthesis, yet the reported efficiency remains unsatisfying (<12 µM/h, see Table S1) due to severe charge recombination, even in the presence of sacrificial agents[18,19]. We synthesized monoclinic Mo:BiVO₄ and anchored $CoO_x$ onto the oxidative {110} facet via photooxidation, which served to promote the water oxidation kinetics. In the meantime, Pd was anchored onto the reductive {010} facet via photoreduction and served to steer the oxygen-reduction pathway from four-electron processes for $H_2O$ formation to two-electron processes for $H_2O_2$ generation. In-depth time-resolved spectroscopic investigations of photocarriers demonstrates that $CoO_x$ and Pd depositions tailored the energetics of the respective facet for improved charge separation, a key obstacle limiting the performance of inorganic photocatalysts. Without using any sacrificial reagent, the reasonably designed $CoO_x$/Mo:BiVO₄/Pd produced $H_2O_2$ at a rate of 1425 µM/h, an apparent quantum yield (AQY) of 1.2% over the full spectrum of sunlight, and a STH of 0.29%, surpassing other inorganic photocatalysts by one order of magnitude (Table S1). With comparable efficiency with organic ones in photocatalytic $H_2O_2$ generation, our work demonstrates the feasibility of applying robust inorganic particulate photocatalysts to efficient photocatalytic $H_2O_2$ generation through delicate design of catalyst spatial and electronic structures.

## Results and discussion

**Synthesis and characterization of $CoO_x$/Mo:BiVO₄/Pd.** We first prepared faceted Mo:BiVO₄ particles using a solid-liquid-reaction method[19,20]. Mo was doped to the V site to increase the bulk conductivity. Mo doping amount was optimized to be 0.025 mol% based on the activity of photocatalytic $H_2O_2$ generation over $CoO_x$/Mo:BiVO₄/Pd (Fig. S2). The X-ray diffraction (XRD) pattern of Mo:BiVO₄ as well as $CoO_x$/Mo:BiVO₄, Mo:BiVO₄/Pd and $CoO_x$/Mo:BiVO₄/Pd particles matched well with that of monoclinic BiVO₄, with {010} and {110} facet peaks located at 30.6° and 18.7°, respectively (Fig. S3). The Brunner−Emmet−Teller (BET) tests show Mo:BiVO₄ as well as $CoO_x$/Mo:BiVO₄, Mo:BiVO₄/Pd and $CoO_x$/Mo:BiVO₄/Pd particles exhibit similar surface areas (1.43–1.71 m²/g, Table S2). The Mo:BiVO₄ particles

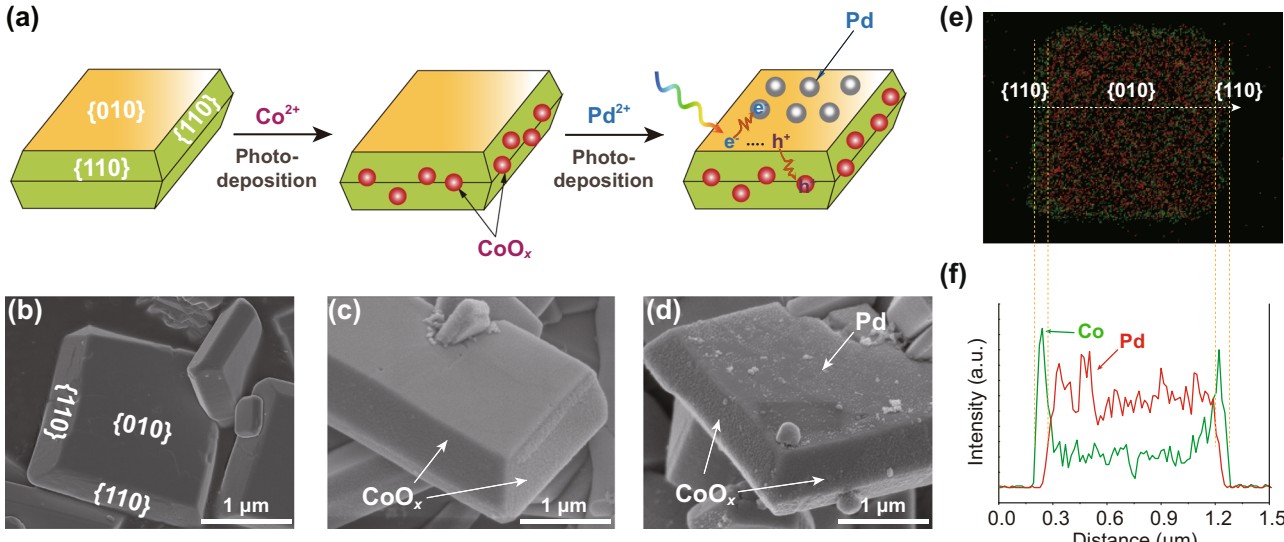

**Fig. 1 Facet-selective loading of $CoO_x$ and Pd cocatalysts on Mo:BiVO₄. a** Schematic deposition processes of $CoO_x$ and Pd on Mo:BiVO₄ and the corresponding SEM images of **b** Mo:BiVO₄, **c** $CoO_x$/Mo:BiVO₄, and **d** $CoO_x$/Mo:BiVO₄/Pd. **e, f** Energy-dispersive X-ray spectroscopy (EDS) elemental mapping and line profile along with the white arrow of $CoO_x$/Mo:BiVO₄/Pd.

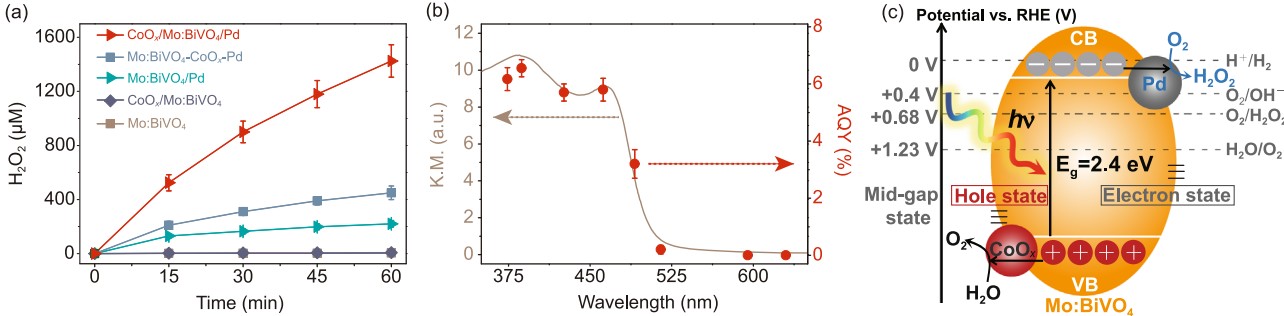

**Fig. 2 Photocatalytic H$_2$O$_2$ generation activities. a** Time courses of photocatalytic H$_2$O$_2$ generation over CoO$_x$/Mo:BiVO$_4$/Pd, Mo:BiVO$_4$-CoO$_x$-Pd, CoO$_x$/Mo:BiVO$_4$, Mo:BiVO$_4$/Pd, and Mo:BiVO$_4$. Reaction conditions: photocatalyst amount, 2 mg; reactant solution, 12 mL PBS aqueous solution (pH = 7.4) saturated with O$_2$; light source, xenon lamp solar simulator, 100 mW/cm$^2$, AM 1.5 G. We note the data series for CoO$_x$/Mo:BiVO$_4$ (7.5 μM at the reaction time of 60 min) overlap with those of Mo:BiVO$_4$ (4.1 μM at the reaction time of 60 min). **b** Apparent quantum yield (AQY) of photocatalytic H$_2$O$_2$ generation over CoO$_x$/Mo:BiVO$_4$/Pd as a function of the incident light wavelength. Reaction conditions: photocatalyst amount, 2 mg; reactant solution, 5 mL PBS aqueous solution (pH = 7.4) saturated with O$_2$; light source, monochromatic LED light. **c** Schematic of photocatalytic H$_2$O$_2$ generation over CoO$_x$/Mo:BiVO$_4$/Pd. CB and VB are short for conduction band and valence band, respectively.

exhibit a decahedron structure with clear facets as shown in scanning electron microscope (SEM) images (Fig. 1a, b). The selected area electron diffraction (SEAD, Fig. S4) pattern confirms the Millier index of top and side facets are {010} reduction facet and {110} oxidation facet, respectively[21]. The Mo doping amount (Mo/V) is 0.023 mol% tested by inductively coupled plasma mass spectrometry (ICP-MS).

Secondly, we selectively loaded cocatalysts onto different facets of Mo:BiVO$_4$ particles via stepwise photodeposition (Fig. 1a). CoO$_x$ as a cocatalyst for water oxidation reaction (WOR) was selectively deposited onto the {110} facet of Mo:BiVO$_4$ via photooxidation of Co$^{2+}$ ions. The loading amount of Co was optimized to be 0.2 wt% based on the activities of photocatalytic H$_2$O$_2$ generation over CoO$_x$/Mo:BiVO$_4$/Pd (Fig. S5). Prominent Co 2p X-ray photoelectron spectroscopy (XPS) peaks demonstrate the successful loading of Co species (Fig. S6a). The Co 2p$_{3/2}$ peak can be deconvoluted to a Co$^{2+}$ peak at 781.6 eV and a Co$^{3+}$ peak at 780.6 eV, suggesting that the valence of Co was in-between +2 and +3, therefore the cocatalyst is denoted as CoO$_x$. The SEM image (Fig. 1c) shows that CoO$_x$ particles are uniformly distributed across the {110} facet of Mo:BiVO$_4$. Consistent with the SEM results, energy-dispersive X-ray spectroscopy (EDS) elemental mapping and line profile (Fig. 1e, f) show 4.1-fold stronger Co signal on the {110} facet compared to that on the {010} facet. Further, SEM (Fig. S7) and TEM (Fig. S8) line profiles of CoO$_x$/Mo:BiVO$_4$ indicate that Co signal on {110} facet is much higher than that on {010} facet. These results demonstrate the selective deposition of CoO$_x$ on the {110} facet of Mo:BiVO$_4$.

Loading CoO$_x$ cocatalyst significantly enhanced the WOR surface kinetics of Mo:BiVO$_4$. The photocatalytic O$_2$ evolution activity of CoO$_x$/Mo:BiVO$_4$ was more than twice as much as that of Mo:BiVO$_4$ (Fig. S9a). The enhancement water oxidation was further verified by the improved photoelectrochemical performance of CoO$_x$/Mo:BiVO$_4$ electrode compared to that of Mo:BiVO$_4$ electrode. The onset potential of the CoO$_x$/Mo:BiVO$_4$ photoanode was ~0.1 V more negative than that of Mo:BiVO$_4$ (Fig. S9b). At a given potential, the photoanodic current density of CoO$_x$/Mo:BiVO$_4$ was much higher than that of Mo:BiVO$_4$ (Fig. S9b). Such stark contrasts between O$_2$ production and photoelectrochemical performance clearly demonstrate the improved WOR activity of Mo:BiVO$_4$ upon CoO$_x$ deposition.

Following the photodeposition of CoO$_x$, Pd as an oxygen-reduction reaction (ORR) cocatalyst was selectively deposited on the {010} facet of Mo:BiVO$_4$ via photoreduction of PdCl$_4^{2-}$ (Fig. 1a). The loading amount of Pd was optimized to be 0.4 wt%

based on the activity of photocatalytic H$_2$O$_2$ generation over CoO$_x$/Mo:BiVO$_4$/Pd (Fig. S10). The distinct Pd 3d XPS peaks indicate the successful loading of Pd species (Fig. S6b). The Pd 3d$_{5/2}$ peak can be deconvoluted to a main Pd$^0$ peak at 335.1 eV and a minor Pd$^{2+}$ peak at 337.0 eV, attributing to the metallic Pd from photoreduction and PdO from partial oxidation of Pd in air, respectively. SEM images (Fig. 1d and S11) show that Pd particles were uniformly and selectively distributed across the {010} facets of Mo:BiVO$_4$. The facet-selective loading of Pd was further demonstrated by the stark contrast between distinctive Pd signal on the {010} facet and negligible Pd signal on the {110} facet from EDS elemental mapping and line profile of Mo:BiVO$_4$ (Fig. 1e, f and S12). SEM (Fig. S13) and TEM (Fig. S14) line profile of Mo:BiVO$_4$/Pd also suggest that Pd signal on {010} facet is much higher than that on {110} facet.

Loading Pd cocatalyst significantly enhanced the selectivity of H$_2$O$_2$ generation from 13% by pristine Mo:BiVO$_4$ to 89% (Fig. S15; H$_2$O$_2$ generation selectivity is defined as the ratio of electrons utilized for H$_2$O$_2$ synthesis to the total number of electrons consumed). These results indicate that Pd, as a verified catalyst for selective H$_2$O$_2$ synthesis[22–24], steered the ORR on Mo:BiVO$_4$ from four-electron (O$_2$ + 4H$^+$ + 4e$^-$ → 2H$_2$O) to two-electron (O$_2$ + 2H$^+$ + 2e$^-$ → H$_2$O$_2$) processes. Moreover, the enhanced H$_2$O$_2$ production selectivity with Pd loading was slightly perturbed by the presence of CoO$_x$, likely attributed to improved H$_2$O$_2$ decomposition (Fig. S15). Note that the two-electron H$_2$ evolution (2H$^+$ + 2e$^-$ → H$_2$), another major side reaction likely limiting the selectivity for H$_2$O$_2$ generation[19], was prohibited because the conduction band of Mo:BiVO$_4$ is too deep to evolve H$_2$.

**Photocatalytic performance**. The photocatalytic H$_2$O$_2$ generation performance of the particulate photocatalyst was evaluated under simulated sunlight irradiation without any sacrificial reagent (Fig. 2a). Bare Mo:BiVO$_4$ exhibits minimal performance for 60-min H$_2$O$_2$ generation (4.1 μM). Loading CoO$_x$ cocatalyst enhanced the photocatalytic H$_2$O$_2$ generation performance by a factor of 1.8 (7.5 μM, Fig. 2a), attributing to promoted water oxidation and consequentially reduced detrimental charge recombination (Fig. 2c). In the meanwhile, loading Pd cocatalyst onto Mo:BiVO$_4$ improved H$_2$O$_2$ generation selectivity (Fig. 2c), resulting in a 53.7-fold enhancement of 60-min H$_2$O$_2$ generation (220.0 μM, Fig. 2a).

Simultaneous loading of CoO$_x$ and Pd cocatalysts enhanced H$_2$O$_2$ generation by 347.6-fold compared to pristine Mo:BiVO$_4$ (Fig. 2a).

Without any sacrificial reagent, $CoO_x$/Mo:$BiVO_4$/Pd generated 1425 μM $H_2O_2$ after one-hour reaction. When the suspension was $N_2$-purged, the $H_2O_2$ production was inhibited by over 99% (Fig. S16), confirming that $H_2O_2$ generation proceeded mainly through ORR. The wavelength-dependent AQYs measured by light-emitting diode (LED) light irradiation agree well with the absorption spectrum of $CoO_x$/Mo:$BiVO_4$/Pd (Fig. 2b), suggesting that $H_2O_2$ was generated following its band-gap excitation. The AQY at 420 nm was determined to be 5.8%, the highest reported for inorganic semiconductors to the best of our knowledge (see Table S1). Furthermore, the AQY of $CoO_x$/Mo:$BiVO_4$/Pd reached 1.2% over the full spectrum, and its STH reached 0.29%. Such an efficiency surpasses other inorganic semiconductor photocatalysts by one order of magnitude (Table S1) and indicates that inorganic photocatalysts are competent for efficient photocatalytic $H_2O_2$ generation. Most importantly, in stark contrast to the high •OH-susceptibility of organic photocatalyst, after 24-h incubation under •OH-rich conditions, $CoO_x$/Mo:$BiVO_4$/Pd exhibits nominal change in $H_2O_2$ productions (Fig. S1b) and chemical compositions (Fig. S17), demonstrating its high resistance to •OH-mediated oxidation.

The photocatalytic $H_2O_2$ generation activity of $CoO_x$/Mo:$BiVO_4$/Pd decreased gradually as shown in Fig. 2a. The reaction rate in the fourth 15 min corresponds to 47% of that in the first one. To clarify such decay, $CoO_x$/Mo:$BiVO_4$/Pd was tested with cycles of reaction. The photocatalytic $H_2O_2$ generation rate in the second cycle was 420 μM/h, 29% of that in the first cycle (Fig. S18a). It is supposed that the decay is related to the gradual transformation of $CoO_x$ to CoPi in $PO_4^{3-}$ solution (applied as $H_2O_2$ stabilizer[25]). Although CoPi has been widely applied to photoanodes a cocatalyst, its roles in photocatalysis are controversial mostly because of inconsistent sample conditions[26]. Here we believe that CoPi solely facilitated the surface kinetics for $O_2$ evolution and hardly affected the energetics for charge separation[27]. Since the photocatalytic $H_2O_2$ generation activity is determined by the charge-separation efficiency, CoPi behaved inferior to $CoO_x$ in our system. The transformation of $CoO_x$ to CoPi was confirmed by characterizing $CoO_x$/Mo:$BiVO_4$/Pd before and after the reaction, where a prominent peak at 133.6 eV attributing to Co-Pi bond was observed in the post-catalysis P $2p_{3/2}$ XPS spectra (Fig. S19). Furthermore, the photocatalytic $H_2O_2$ generation activity of freshly prepared CoPi/Mo:$BiVO_4$/Pd was 21% of that of $CoO_x$/Mo:$BiVO_4$/Pd and was similar to that of spent $CoO_x$/Mo:$BiVO_4$/Pd (Fig. S20). These results confirm that the transformation of $CoO_x$ to CoPi undermined $CoO_x$/Mo:$BiVO_4$/Pd for photocatalytic $H_2O_2$ generation. In order to avoid the deterioration of $CoO_x$/Mo:$BiVO_4$/Pd in $PO_4^{3-}$ solution, photocatalytic $H_2O_2$ generation was conducted in pure water as shown in Fig. S18b. As expected, $CoO_x$/Mo:$BiVO_4$/Pd was highly stable over five cycles of reaction. To further demonstrate the stability of $CoO_x$/Mo:$BiVO_4$/Pd, we test the performance of $CoO_x$/Mo:$BiVO_4$/Pd in seawater, which is a desirable solution condition for artificial photosynthesis[28]. No deactivation was observed over five-round repetitive use (Fig. S21). Yet the cumulative production of $H_2O_2$ was lower than that in phosphate solution owing to $H_2O_2$ decomposition, consistent with the previous report[29]. In following studies, the cumulative production of $H_2O_2$ in pure water will be improved with rapid $H_2O_2$ diffusion by a large-scale photosynthesis setup where the $CoO_x$/Mo:$BiVO_4$/Pd photocatalyst will be immobilized on a support using drop-casting or screen printing technologies and integrated in a flow cell photolysis system[19,30].

**Charge separation**. We note that the enhancement of $H_2O_2$ production by the synergistic effect of coloading Pd and $CoO_x$ is even higher than the multiplication of the enhancements by loading Pd and $CoO_x$ individually, i.e., 347.6-fold for coloading Pd and $CoO_x$, 53.7-fold for solely loading Pd, and 1.8-fold for solely loading $CoO_x$. In the meantime, when $CoO_x$ and Pd were randomly deposited on Mo:$BiVO_4$ (denoted as Mo:$BiVO_4$-$CoO_x$-Pd with a SEM image in Fig. S22), it was only 32% as active as $CoO_x$/Mo:$BiVO_4$/Pd, though these two catalysts had similar surface kinetics and selectivity. Further, the photocatalytic $O_2$ evolution activity of $CoO_x$/Mo:$BiVO_4$/Pd was 3.2-fold higher than that of $CoO_x$/Mo:$BiVO_4$ (Fig. S9a). These results suggest that selective coloading of $CoO_x$ and Pd not only improved surface kinetics for $O_2$ evolution and selectivity for $H_2O_2$ production as introduced above, but also tuned other critical processes like charge separation[31]. Furthermore, selectively coloading dual cocatalysts has been applied to enhance charge separation for sacrificial photocatalytic $O_2$ evolution[21,32,33]. To this end, charge separation processes in Mo:$BiVO_4$, $CoO_x$/Mo:$BiVO_4$, Pd/Mo:$BiVO_4$, $CoO_x$/Mo:$BiVO_4$/Pd were thoroughly studied by transient absorption spectroscopy (TAS).

The TA spectra of photogenerated charge carriers in Mo:$BiVO_4$ was examined upon band-gap excitation as shown in Fig. S23. The spectra exhibited strong absorption in 20,000–17,000 cm$^{-1}$, a broad absorption in 170,000–5000 cm$^{-1}$, and a weak absorption <5000 cm$^{-1}$. These absorptions are attributed to trapped holes, deeply trapped electrons and free/shallowly trapped electrons, respectively (see supplementary discussions in Figs. S23 for detailed peak assignments)[34–36]. The dynamics of these photocarriers were compared in the ultrafast region (picosecond time-scale) as shown in Fig. 3. The photocarriers probed at 781 nm (~1.59 eV, deeply trapped electrons) and 505 nm (~2.45 eV, trapped holes) exhibited comparably slow decay kinetics, suggesting that these photocarriers recombined non-radiatively. In contrast, the photocarriers probed at 5000 nm (the free/shallowly trapped electrons) exhibited much faster decay kinetics, which well explains the weak absorption <5000 cm$^{-1}$ shown in Fig. S23. This result indicates that the free/shallowly trapped electrons were rapidly trapped by the mid-gap states in Mo:$BiVO_4$, consistent with previous studies[35,36]. Regardless of such an unfavorable effect, the TA signals for this kind of photocarriers were still observed to vary significantly after loading Pd or $CoO_x$ on Mo:$BiVO_4$ (Fig. 3b). Loading Pd on the {010} facet of Mo:$BiVO_4$ accelerated the decay of the free/shallowly trapped electrons since Pd captured electrons. In contrast, loading $CoO_x$ on the {110} facet lead to an opposite effect because $CoO_x$ captured holes and thus increased the electron population in Mo:$BiVO_4$. At 50 ps, for instance, loading $CoO_x$ increased the intensity of TA signal for free/shallowly trapped electrons by ~66%, while loading Pd only decrease the intensity by ~6% compared to that of bare sample. The effect of $CoO_x$ on the dynamics of the free/shallowly trapped electrons was more intense than that of Pd because electron transfer to Pd competed with trapping of electrons into deep trap states below the CB. Electron trapping to deep trap states was represented by the strong TA signal probed at 781 nm as shown in Fig. 3a.

Considering the crucial impact of electron trapping on electron-transfer processes in the picosecond region, the electron dynamics was further examined in the microsecond-millisecond region where photoexcited electrons in Mo:$BiVO_4$ are expected to have already relaxed in the trap states. As shown in Fig. 4a, Pd and $CoO_x$ accelerated and decelerated the decay of free/shallowly trapped electrons, respectively, similar to the effects in picosecond region (Fig. 3b). This result indicates that in this time region, Pd and $CoO_x$ captured electrons and holes, respectively, as expected. The population of free/shallowly trapped electrons that remained in Mo:$BiVO_4$ after electrons are captured by Pd can be

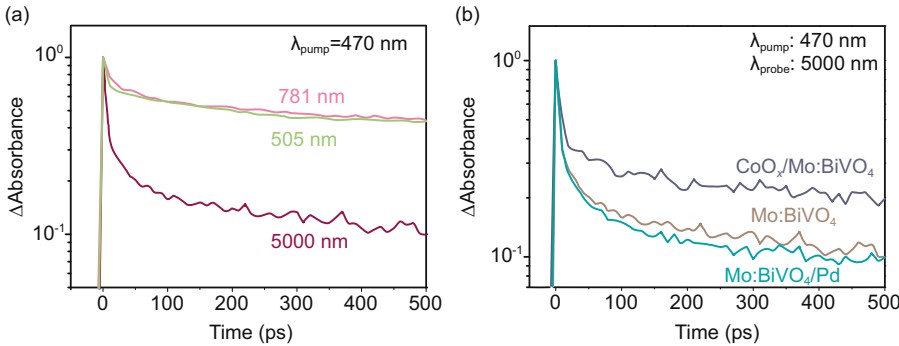

**Fig. 3 Charge-carrier dynamics at picosecond time-scale. a** Transient profiles of photocarriers probed at 505 nm (trapped holes), 781 nm (deeply trapped electrons), and 5000 nm (free/shallowly trapped electrons) in Mo:BiVO$_4$. **b** Transient profiles of photocarriers probed at 5000 nm (free/shallowly trapped electrons) for Mo:BiVO$_4$, CoO$_x$/Mo:BiVO$_4$ and Mo:BiVO$_4$/Pd. Pump wavelength: 470 nm (4 μJ pulse$^{-1}$).

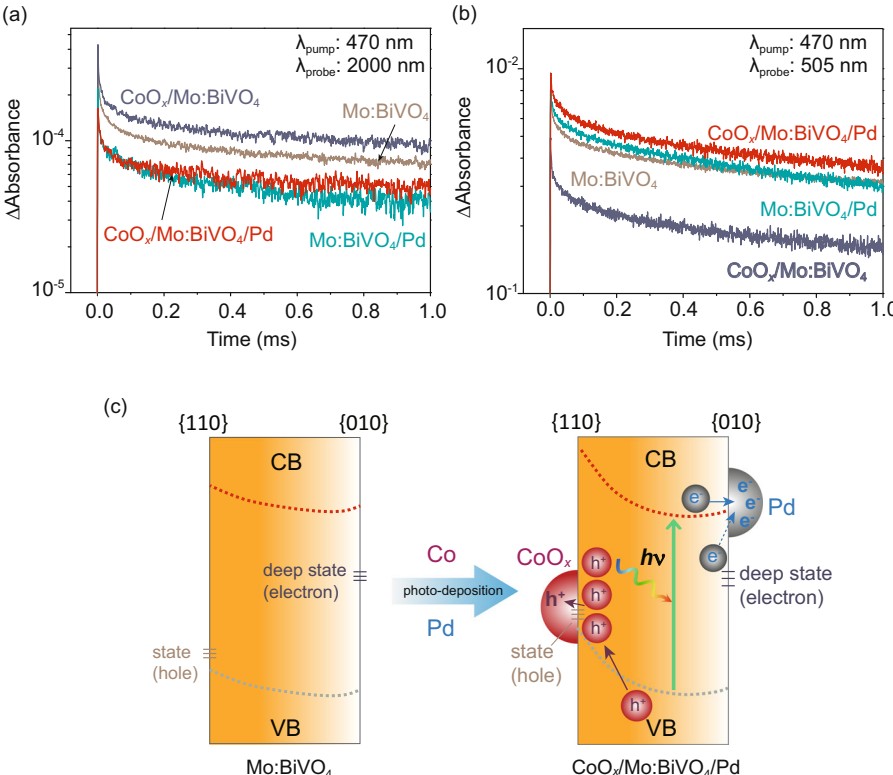

**Fig. 4 Charge-carrier dynamics at microsecond time-scale and impacts of cocatalysts on the energetics of Mo:BiVO$_4$ facets.** Transient profiles of photocarriers probed **a** at 2000 nm (free/shallowly trapped electrons) and **b** 505 nm (trapped holes) for Mo:BiVO$_4$, CoO$_x$/Mo:BiVO$_4$, Mo:BiVO$_4$/Pd, and CoO$_x$/Mo:BiVO$_4$/Pd. Samples were excited by 470 nm laser pulses (Surelite I, duration: 6 ns, fluence: 3 mJ pulse$^{-1}$, repetition: 1 Hz). **c** Impacts of cocatalysts on the energetics of Mo:BiVO$_4$ facets.

determined by estimating the ratio of ΔAbsorbance displayed in Fig. 4a with respect to bare Mo:BiVO$_4$ and CoO$_x$/Mo:BiVO$_4$ for Mo:BiVO$_4$/Pd and CoO$_x$/Mo:BiVO$_4$/Pd, respectively. At 200 μs, for instance, 59% and 42% of free/shallowly trapped electrons remained in Mo:BiVO$_4$ for Mo:BiVO$_4$/Pd and CoO$_x$/Mo:BiVO$_4$/Pd, respectively. This implies that 41 and 58% of free/shallowly trapped electrons transferred to Pd for Mo:BiVO$_4$/Pd and CoO$_x$/Mo:BiVO$_4$/Pd, respectively. More electrons transferred to Pd on CoO$_x$/Mo:BiVO$_4$/Pd than on Mo:BiVO$_4$/Pd because of more efficient charge separation (synergistic charge separation) in the former case. To further justify this effect, the impact of CoO$_x$ and Pd on the decay kinetics of trapped holes in Mo:BiVO$_4$ was investigated. To further justify this effect, the impact of CoO$_x$ and Pd on the decay kinetics of trapped holes in Mo:BiVO$_4$ was

investigated. As depicted in Fig. 4b, CoO$_x$ and Pd accelerated and decelerated the decay of accumulated trapped holes, respectively. Surprisingly, in CoO$_x$/Mo:BiVO$_4$/Pd, the effect of CoO$_x$ on capturing photogenerated holes in Mo:BiVO$_4$ was compensated by the effect of Pd on accumulating trapped holes by efficiently trapping electrons. Furthermore, CoO$_x$/Mo:BiVO$_4$/Pd accumulated more trapped holes in Mo:BiVO$_4$, even higher than that in Mo:BiVO$_4$/Pd. This finding reveals that electrons and holes were efficiently separated at different facets. Aside from accumulating photogenerated holes in Mo:BiVO$_4$ and facilitating the transfer of free/shallowly trapped electrons to Pd, loading cocatalysts also activated the deeply trapped electrons in Mo:BiVO$_4$ for photocatalysis. As shown in Fig. S24, the TA signal and decay kinetics of the deeply trapped electrons in CoO$_x$/Mo:BiVO$_4$/Pd were

similar to that of Pd/Mo:BiVO$_4$ and apparently different from that of Mo:BiVO$_4$ and CoO$_x$/Mo:BiVO$_4$. This result suggests that the deeply trapped electrons transferred to Pd and became available for subsequent surface reactions in Mo:BiVO$_4$/Pd and CoO$_x$/Mo:BiVO$_4$/Pd. The charge transfer involving deeply trapped electrons to Pd cocatalyst is possible via tunneling and trap-to-trap hopping. A similar diffusion of trapped electrons is often proposed to take place on long-lived persistent phosphor materials[37], wherein the trap states are situated at ~0.5–1.0 eV below the conduction band minimum. Aside from this diffusion process, re-excitation of deeply trapped electrons to the CB and eventually transfer to the Pd will be possible. Taken together, the above findings demonstrate that selectively coloading Pd and CoO$_x$ on the expected facets of Mo:BiVO$_4$ significantly enhanced the charge separation and suppressed rapid charge-carrier trapping and recombination. Such positive effects are supposed to be achieved by tuning the energetics between cocatalysts and respective Mo:BiVO$_4$ facet (Fig. 4c and S25).

In conclusion, we developed an inorganic semiconductor-based system for efficient overall photocatalytic H$_2$O$_2$ generation. Faceted Mo:BiVO$_4$ particles was used as a light absorber and its {110} and {010} facets were selectively loaded with CoO$_x$ and Pd as WOR and ORR cocatalysts, respectively. These cocatalysts in such a configuration greatly improved the kinetics and selectivity for surface reactions. Furthermore, the spatial separation of cocatalysts on different facets of Mo:BiVO$_4$ significantly enhanced the charge separation and suppressed rapid charge-carrier trapping and recombination, a key challenge in improving the efficiency of inorganic photocatalysts. With these merits, CoO$_x$/Mo:BiVO$_4$/Pd generated H$_2$O$_2$ with an AQY of 1.2% at full spectrum and a STH of 0.29%, a new record for inorganic semiconductor-based systems.

## Methods

**Catalyst preparation.** Single crystal Mo:BiVO$_4$ was prepared by heating the mixture of K$_2$CO$_3$ (1.047 g), MoO$_3$ (1.8 mg) and V$_2$O$_5$ (2.272 g) in a ceramic crucible at a heating rate of 1.5 °C/min to 450 °C and annealing for 5 h in a muffle furnace. The obtained Mo:K$_3$V$_5$O$_{14}$ (2 g) was mixed with Bi(NO$_3$)$_3$•5H$_2$O (0.326 g) and dispersed in 50 mL deionized water under ultrasonication for 30 min. The mixture was stirred and heated at 70 °C for 10 h under ultrasonication, separated by centrifugation, washed with deionized water, and dried at 70 °C for 8 h. As-prepared Mo:BiVO$_4$ (0.2 g) was dispersed in 100 mL water, followed by addition of 0.1 mol NaIO$_3$ and 0.27 mL Co(NO$_3$)$_2$ stock (1.5 g/L). The mixture was irradiated at λ > 420 nm for 3 h using a xenon lamp solar simulator (model 300 DUV; Perfect Light, Inc., light intensity = 0.1 W/cm$^2$), filtered, washed with deionized water, and dried at 60 °C for 8 h. The as-prepared CoO$_x$/Mo:BiVO$_4$ (0.15 g) was dispersed in 100 mL pure water, followed by addition of 0.18 mL Na$_2$PdCl$_4$ stock (3.3 g/L). The mixture was irradiated at λ > 420 nm for 3 h. As-prepared CoO$_x$/Mo:BiVO$_4$/Pd was filtered, washed with deionized water, and dried at 60 °C for 8 h. Same photo-deposition method was also applied in preparing Mo:BiVO$_4$/Pd except for using Mo:BiVO$_4$ as the starting material. Mo:BiVO$_4$-CoO$_x$-Pd was prepared following an impregnation procedure by air-purging the mixture of Mo:BiVO$_4$ (0.2 g), Co(NO$_3$)$_2$ (0.4 mg) and Na$_2$PdCl$_4$ (0.8 mg) (from stock) until dry, followed by heating in a ceramic crucible at a heating rate of 5 °C/min to 200 °C and annealing for 0.5 h under reductive condition (10% H$_2$ and 90% Ar) in a tube furnace. The elemental compositions were analyzed by EDS, XPS, and ICP-MS (Figs. S26 and 27 and Table S3).

**Photocatalyst characterizations.** XPS measurements were performed with a Thermo Scientific 250Xi system with monochromatic Al Kα as the excitation source. The XRD patterns were recorded with a Bruker D8 Advance X-ray diffractometer with Cu Kα radiation (λ = 1.5406 Å) operated at 40 kV and 40 mA. The BET tests were performed by an ASAP 2460 with N$_2$ analysis adsorptive at 77.2 K. SEM images were taken with a Hitachi SU-8010 microscope equipped with EDS at 30 kV. TEM images were taken using a Hitachi 7650 microscope operated at 100 kV. UV-DRS spectra was taken with a Shimadzu UV-3600 with a resolution of 0.1 nm. The ICP-MS measurement was performed with a NexION 300X (detection limit 1 μg/L).

**Photocatalytic activity tests.** Photocatalyst (24 mg) was dispersed in 12 mL deionized water with 1 M phosphate buffer (pH 7.4) in a custom-made reactor containing a quartz window. The catalyst was dispersed by ultrasonication for

10 min and purged with O$_2$ for 20 min. All the equipment needed is shown in Fig. S28. Photocatalytic production of H$_2$O$_2$ was assessed by irradiating photo-catalyst suspension using a xenon lamp solar simulator (model 300 DUV; Perfect Light Inc.) under water bath (12 ± 0.5 °C). The light intensity was adjusted to 100 mW/cm$^2$ (AM 1.5 G; irradiation area = 1.83 cm$^2$). The Xenon lame and the standard AM1.5 G (ASTMG 173) spectrum is shown in Fig. S29. For the wavelength-dependent AQY analysis, the photolysis was performed using LED light irradiation (model slight; Perfect Light, Inc.). At designated time points, 50 μL suspension was taken for analysis of H$_2$O$_2$ productions and diluted with phosphate buffer (pH = 7.4) to a H$_2$O$_2$ concentration (2–20 μM) that is most suitable for accurate H$_2$O$_2$ quantification, followed by centrifugation. Then 50 μL supernatant was taken and mixed with 50 μL solutions containing phosphate buffer (50 mM, pH = 7.4), amplifu red (100 μM) and horseradish peroxidase (0.05 U/mL). Amplifu red selectively reacted with H$_2$O$_2$ in the presence of horseradish peroxidase and formed the product resorufin. Resorufin in the mixture solution was quantified using an Agilent high-performance liquid chromatography coupled to a photo-diode array detector (detection at 560 nm); 50 μL of each sample was injected. The calibration in Fig. S30 is used to quantitatively analyze the H$_2$O$_2$ concentration. Separation was carried out in a C18 column at 20 °C with an iso-cratic mobile phase of 55% sodium citrate buffer (with 10% methanol (v/v), pH 7.4) and 45% methanol (v/v) at a flow rate of 0.5 mL min$^{-1}$. The AQY was determined using:

$$AQY = \frac{2 \times [H_2O_2]_{1h} \times V}{I_{tot\_p} \times A \times t} \tag{1}$$

where [H$_2$O$_2$]$_{1h}$ is the H$_2$O$_2$ concentration, $I_{tot\_p}$ is the total photo flux of simulated sunlight irradiation (4.4 × 10$^{-3}$ mol/m$^2$/s, calculation details were shown in section S3), $V$ is the volume of suspension (12 mL), $A$ is the irradiation area (1.91 cm$^2$ in this study), and $t$ is the reaction time (1 h).

The STH was determined using:

$$STH = \frac{\triangle G(H_2O_2) \times [H_2O_2]_{1h} \times V}{I_{tot\_e} \times A \times t} \tag{2}$$

Where $\Delta G$(H$_2$O$_2$) is the free energy for H$_2$O$_2$ formation (117 kJ mol$^{-1}$), $I_{tot\_e}$ is the total intensity of simulated sunlight irradiation (0.1 W/cm$^2$), $V$ is the volume of suspension (12 mL), $A$ is the irradiation area (1.91 cm$^2$ in this study), and $t$ is the reaction time (1 h).

**Photoelectrochemical characterizations.** Particle-based electrodes was prepared by a particle transfer method[38]. Firstly, 10 mg prepared CoO$_x$/Mo:BiVO$_4$ particles were suspended in a 450 μl isopropanol, followed by sonicated for 5 min. Secondly, the uniform suspension solution was dropped casting on a 1 × 3 cm glass substrate and fully dried in air. Thirdly, a thin layer of Ti (2–5 nm) was sputtered on the CoO$_x$/Mo:BiVO$_4$ particles. At last, the transferred electrode was sonicated for 10 s in water to remove the excessive particles on the surface. The electrochemical properties were assessed on a Biologic SP150 electrochemical analyzer using a three-electrode cell with the as-prepared electrode as the working electrode, Ag/AgCl as the reference electrode, and glassy carbon as the counter electrode. Cyclic voltammetry curves were obtained in a N$_2$- or O$_2$-saturated phosphate buffer solution (0.5 M, pH = 6.5). All potentials versus Ag/AgCl were converted to values vs. RHE.

**Photocarriers dynamics by TAS measurement.** Microsecond-millisecond TAS characterizations were performed using Nd:YAG laser system (Continuum, Surelite I) equipped with custom-built spectrometers[39,40]. Briefly, the TA spectra was measured from 20,000 cm$^{-1}$ (500 nm)–1600 cm$^{-1}$ (6250 nm) after band-gap excitation using 470 nm laser pulses (duration: 6 ns, fluence: 3 mJ pulse$^{-1}$). For the IR probing, the light emitted from the MoSi$_2$ coil was focused on the sample and then the reflected light from the sample was introduced to a grating spectrometer. The photoexcited electrons was probed at 2000 cm$^{-1}$ (2000 nm). The mono-chromated light was detected by a mercury cadmium telluride (MCT) detector (Kolmar). Meanwhile, the photogenerated trapped electrons and holes in the non-modified and (CoO$_x$, Pd)-modified Mo:BiVO$_4$ were probed at 12,800 cm$^{-1}$ (781 nm) and 19,800 cm$^{-1}$ (505 nm, 2.45 eV), respectively. The output electric signal was amplified with an AC-coupled amplifier (Stanford Research Systems, SR560, 1 MHz). The time resolution of the spectrometer was limited to 1 μs by the response of the MCT detector. The output electric signal was amplified using AC-coupled amplifier with a bandwidth of 1 MHz, which measures responses from one microsecond to milliseconds. Three thousand responses were accumulated to obtain the intensity trace at one particular wavenumber or probe energy. The experiments were carried out in vacuum and at room temperature.

Femtosecond time-resolved absorption measurements were performed by employing a pump-probe technique based on femtosecond Ti:Sapphire laser system (Spectra Physics, Solstice & TOPAS prime; duration = 90 fs; repetition rate = 1 kHz)[40]. The time resolution of this spectrometer was ~90 fs. Briefly, in this experiment, the photoexcited charge carriers in the photocatalysts were probed at 19,800 cm$^{-1}$ (505 nm), 12,800 cm$^{-1}$ (781 nm), and 2000 cm$^{-1}$ (5000 nm). In the mid-IR absorption measurement, the probe light transmitted from the sample was detected by an MCT detector (Kolmar), while in the visible to near-infrared region,

the diffuse-reflected probe light was detected by photomultiplier (Hamamatsu Photonics, H11903-20). The samples were excited by 470-nm pulses (duration: 90 fs, fluence: 4 μJ pulse$^{-1}$). To obtain the absorbance change with a good signal-to-noise ratio, the pump pulses were chopped using an optical chopper at 500 Hz and the signal acquisition was carried out on a shot-by-shot basis at a rate of 1 kHz. The decay curves were obtained at 10 ps intervals and accumulated signals were averaged over 1000–4000 scans for one point. TAS measurements were performed in vacuum (base pressure ∼ 10$^{-5}$ Torr). For sample preparation, each Mo:BiVO$_4$ and (Pd, CoO$_x$)-loaded Mo:BiVO$_4$ powders was prepared by dispersing the powder on isopropanol and then drop-casted on a circular CaF$_2$ substrate and subsequently dried naturally in air to obtain a powder film with a density of ∼ 1.25 mg cm$^{-2}$.

## Data availability

Source data are provided with this paper.

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

## Acknowledgements

This work was supported by National Natural Science Foundation of China (NSFC, No. 22006129), the Fundamental Research Funds for the Central Universities (No. 2020FZZX001-06) and JSPS KAKENHI Grant Number JP20K22556. We are grateful to Sudan Shen (State Key Laboratory of Chemical Engineering at Zhejiang University) and analysis center of agrobiology and environmental sciences for help in TEM and SEM measurements, respectively.

## Author contributions

C.C., Z.P., and K.D. designed research; T.L. and Z.P. synthesized the catalysts and conducted performance test; T.L. conducted SEM, TEM, and XPS measurements; J.J.M.V., K.K., A.Y. performed TAS measurements and analysis. T.L., C.C., Z.P., B.C., K.K., and K.D. analyzed data; T.L., C.C., Z.P., B.C., and K.D. wrote the paper. All authors discussed the results and commented on the manuscript.

## Competing interests

The authors declare no competing interests.

**Additional information**

