## [Peer Review File · Nature Communications]

REVIEWER COMMENTS

Reviewer #1 (Remarks to the Author):

The authors report the design of BiVO₄ with distinct facets with the function of spatial separation of electrons and holes. Based on the redox reaction induced at the specific facet, cocatalysts particles can be selectively deposited on the desired facet of BiVO₄ (i.e. CoOx and Pd). With the Pd as cocatalyst for reduction site, H₂O₂ was generated as a result of oxygen reduction reaction. The authors claim the highest efficiency of AQY (full spectrum) among all reports. Although the results are encouraging and characterisation are decent, this work raises concerns in terms of novelty, innovation as well as some fundamental aspects of the observed results. The reviewer considers this work not at the expected high level for Nature Communications.

1. Selective photodeposition of dual cocatalysts on faceted BiVO₄ or other Bi-based ternary oxides has been widely reported. Selective deposition of CoOx on the oxidative facet {110} and selective deposition of precious metals on reductive facet of {010} have been investigated thoroughly. Following this spatial location of cocatalyst, it performed better than randomly decorated cocatalysts in various redox reactions. In this regard, the novelty and impact are not as high.
2. The authors elaborated that the transformation of CoOx to CoPi within 15 mins of photocatalytic reaction. Although the CoPi is cocatalysing oxygen evolution, the reviewer believes this "deactivation" is significant.
3. The thus-formed H₂O₂ can go through the further oxidative decomposition by BiVO₄. Oxidation of H₂O₂ and water would become competitive reactions. This might also be the reason of non-linear observation of H₂O₂ with time. The authors may want to look into this aspect. If H₂O₂ is not separated from the photocatalytic system upon generation, basically the system may not last long.
4. The authors stated that the reaction is determined by the charge-separation efficiency. However, the reviewer think that it is the charge transfer mechanism that determines the selectivity of ORR to H₂O₂ instead of other radicals. Charges are separated within the photocatalyst but the charge transfer affected the product selectivity.
5. Grammatical errors can be spotted throughout the manuscript: e.g.
 - The TA signals...was still
 - To clarity such decay....
6. How would the deep trapped electrons be able to move to Pd as illustrated in Figure 4c?
7. Efficiencies (AQY and other) are higher but the reaction time of 15 mins are relatively not practical.

Reviewer #2 (Remarks to the Author):

Overall photosynthesis of H₂O₂ by an inorganic semiconductor

T. Liu et al.

First of all, I apologize the delay in reviewing this paper.

The authors report photocatalytic synthesis of hydrogen peroxide from water and oxygen using a BiVO₄:Mo photocatalyst with the {110} and {010} facets modified by CoO_x and Pd, respectively. The reported reaction efficiency is impressive (apparent quantum yield = 1.2%, solar-to-chemical conversion efficiency at full spectrum = 0.29%). Then, I recommend publication of this paper in Nature Communications after minor revision.

Comments

1. Lines 131, 132, 184, 185: The enhancement factors should be expressed by taking the significant figures of the data in Figure 2a.
2. Lines 215-218: This part should be explained more clearly and in more detail.
3. Lines 225-229: These sentences should be rewritten.
4. Lines 238-240: The data for CoO_x/BiVO₄:Mo/Pd is missing in Fig. S17.
5. Lines 240-242: The sentence should be corrected.
6. Fig. 4: I would like to know the origin for the band bending between the {110} and {010} facets. Also, does it exist not only on the surface of BiVO₄ but also in the bulk?

Reviewer #3 (Remarks to the Author):

This manuscript reports on a very nice piece of works dealing with the use of CoOx/BiVO4:Mo/Pd heteromicrostructures for the efficient production of H2O2 by photocatalysis under solar light. First of all, the solar-to-chemical conversion efficiencies reported in this work outperform by one order of magnitude the values reported so far for inorganic photocatalysts. Even though this kind of heterostructures involving faceted BiVO4 microcrystals have already been widely investigated for water reduction reaction and water oxidation reaction, the main interests of manuscript rely on the use of the Mo as dopant to improve the conductivity of BiVO4 microcrystals and on the use of many complementary techniques to determine the origin of the improvements evidenced with these heterostructures. Using a suitable combination of steady-state and transient absorption spectroscopy techniques, the authors were able to provide convincing features about the electronic processes involved in the photocatalytic process. The work follows the highest standard in the field of photocatalysis and the conclusions are well-supported by the experimental data. Consequently the reviewer recommends the publication of this work after minor revisions according to the following comments:

1. The controlled photodeposition of metallic particles or metal oxide particles on BiVO4 microparticles exposing well-defined facets in order to obtain heterostructures has already been reported by different groups. The authors should therefore quote some references about this point, especially the pioneer's work of Can Li et al. (Nat. Commun. 2013, 4, 1432 and Ener. Environ. Sci. 2014, 7, 1369) or more recent ones for efficient water oxidation reactions.
2. A scheme or photograph of the experimental set-up should be shown in the SI file in order to be able to reproduce the experiments. Moreover, the emission spectrum of the Xenon lamp employed should be provided.
3. The actual chemical composition of the heterostructures should be determined more precisely. First of all, the actual presence of Mo was not evidenced. Moreover, the experimental bulk Pd/Bi and Co/Bi atomic ratios should be reported because it seems that the amount reported in the article are nominal loadings. XPS and EDX analyses would provide information about the surface and bulk composition of the materials prepared. That would improve the manuscript.
4. Photocatalytic properties of a material are related to surface properties. As a result the question arises about the evolution of the specific areas when the co-catalysts are deposited. The reviewer would therefore suggest to give the specific surface areas of the different photocatalysts studied and to comment these values. Moreover, what is the influence of the sizes of the Pd and CoOx nanoparticles on the photocatalytic properties? Are the sizes of these nanoparticles reproducible from one batch to another? If not, does it influence the photocatalytic properties?

Reviewer #4 (Remarks to the Author):

In this manuscript, the authors prepared faceted BiVO₄:Mo particles selectively loaded with CoOx and Pd as WOR and ORR cocatalysts on {110} and {010} facets for efficient overall photocatalytic H₂O₂ generation, achieving an AQY of 1.2% at the full spectrum and a STH of 0.29%, a new record for inorganic semiconductor-based systems. However, there are some problems needed to be solved before it can be recommended for publication:

1. The authors only present the XRD pattern of BiVO₄:Mo. After loading cocatalysts onto BiVO₄:Mo, the authors should also provide their XRD patterns.
2. More evidence should be provided for the Facet-selective loading of CoOx and Pd cocatalysts on BiVO₄:Mo, such as high-resolution TEM.
3. Line 121: The authors mentioned that the enhanced H₂O₂ production selectivity with Pd loading was not perturbed by the presence of CoOx. However, according to Figure S10, H₂O₂ production selectivity was perturbed in the presence of CoOx actually. It is suggested that the authors give corresponding explanations based on subsequent analysis.
4. Various comparisons of photocatalytic H₂O₂ generation performance are given elaborately. Since WOR cocatalyst plays an important role in charge separation, the oxygen evolution part of the photocatalytic system should not be ignored. How about the time course of O₂ evolution over CoOx/BiVO₄:Mo/Pd?
5. Line 143-line 146: More evidence should be provided for high resistance to •OH-mediated oxidation of CoOx/BiVO₄:Mo/Pd, such as XPS spectra of BiVO₄ before and after action.
6. Line 163: The authors believed that CoPi solely facilitated the surface kinetics for O₂ evolution and hardly affected the energetics for charge separation. If CoPi had no adverse effects, why the photocatalytic H₂O₂ generation rate of CoOx/BiVO₄:Mo/Pd decreased gradually?

Reviewer #5 (Remarks to the Author):

Review of NCOMMS-21-38618-T

Summary

In this article, titled, "Overall photosynthesis of H₂O₂ by an inorganic semiconductor", the authors develop a robust all-inorganic photocatalyst system (CoOx on {110}/ Mo: BiVO₄/ Pd on {010}) for the photocatalytic synthesis of peroxide from water (STC of ~0.29%). In this article, the authors synthesise nanoparticulate Mo-doped BiVO₄ with distinct {110} and {010} facets. The co-catalysts CoOx and Pd were selective deposited on the {110} and {010} facets, respectively. The physical

properties of the photocatalyst, and its parent materials, were well characterised using XRD, SEM with EDS, TEM & SAED, XPS etc. The photocatalytic activity of the photocatalyst, and its parent materials, were also well characterised, with the activities expressed as apparent quantum yields (AQY) at a fixed wavelength, full spectrum quantum yield and solar-to-chemical conversion efficiency for the selective formation of H₂O₂ from splitting water. Also, the photophysical behaviour of the photocatalyst, and its parent materials, were explored using ultra-fast and slow timescale TAS, which provided evidence for the charge transfer of holes and electrons to the CoO_x and Pd co-catalysts, respectively, and its benefits with respect to increases in charge carrier lifetimes.

Overall, the work is of a very high standard, and I recommend its publication in your journal (minor corrections). Below there are some typos from the Abstract and Introduction sections and some specific queries that I would like addressed before the final submission.

Typos/ grammatical errors

T1) "on proper facet" should be "on distinct facets"

T2) "With regard of reaction" should be "With regard to reaction"

T3) "are advantageous on mass" should be "are advantageous for the mass"

T4) "greatly reduces concentration" should be "greatly reduces the concentration"

T5) "reported efficiency remain unsatisfying" should be "reported efficiency remains unsatisfying"

T6) "due to severe charge recombination even" should be "due to severe charge recombination, even"

T7) "leveraging the record" should be "surpassing the record"

I have only provided typos for the Abstract and Introduction section. I recommend that the entire article is proof read once more before its final submission.

Queries

Q1) I don't know what the convention is for Nature Communications, but from what I have seen, it is more usual to write, "Mo:BiVO₄" than "BiVO₄:Mo" (see Google Ngram Viewer statistics on this). I therefore suggest you use Mo:BiVO₄ throughout.

Q2) In your XPS analysis section, you write "The Pd 3d_{5/2} peak can be deconvoluted to a main Pd⁰ peak at 335.1 eV and a minor Pd²⁺ peak at 337.0 eV, attributing to the metallic Pd from photoreduction and PdO₂ from partial oxidation of Pd in air, respectively."

Do you mean PdO, as the oxidation state of Pd is 2+?

Q3) You achieve an AQY at 420 nm of 5.8%, which is the highest reported for inorganic semiconductors to date. I recommend that this finding be highlighted in the abstract.

Q4) In Figure 2a, the data for BiVO₄:Mo overlaps with that of CoO_x/ BiVO₄:Mo, so it is hard to see. I'm not sure what is the best way to resolve this. Perhaps make one of the data series 50%

transparent so that the other can be seen, or state in the figure caption that data from these two series overlap.

Q5) You write, "The cumulative production of H₂O₂ in pure water will be improved by a large scale photosynthesis setup with rapid H₂O₂ diffusion in a following study." I'm not sure what you mean by rapid diffusion here. Please make this more clear to the reader how the rapid diffusion of H₂O₂ will prevent its decomposition in future work.

Q6) You write, "We note that the enhancement on H₂O₂ production by coloaded Pd and CoO_x is even higher than the multiplication of individual enhancements". Perhaps use the term 'synergistic'.

Q7) In your experimental section, you do not explain if your samples were investigated in air or in solution for your transient absorption spectroscopy studies. Please state the medium in which you samples were measured in the experimental, and also, the figure caption (Figure 4).

Q8) In Figure 4c you show band energies and band bending in your photocatalysts loading with co-catalysts. You also assume that the energy of the {110} facet is more negative than that of the {010} facet. How were these energies determined? (facets, Fermi energies of the co-catalysts, band-bending etc.). Please provide some details of this calculation in your experimental section.

Q9) In your experimental section on "Photocatalytic activity tests", please provide details, if any, on the calibration of your H₂O₂ measurement set-up using known concentrations of H₂O₂ from a stock solution. Please provide this curve, if any, in your supporting information (apologies if this information is already included, and I have missed it).

Response to Comments

Overall photosynthesis of H₂O₂ by an inorganic semiconductor

Tian Liu, Zhenhua Pan, Junie Jhon M. Vequizo, Kosaku Kato, Binbin Wu, Akira Yamakata, Kenji Katayama, Baoliang Chen, Chiheng Chu, Kazunari Domen

Reviewer 1

Comment 1.1 The authors report the design of BiVO₄ with distinct facets with the function of spatial separation of electrons and holes. Based on the redox reaction induced at the specific facet, cocatalysts particles can be selectively deposited on the desired facet of BiVO₄ (i.e. CoO_x and Pd). With the Pd as cocatalyst for reduction site, H₂O₂ was generated as a result of oxygen reduction reaction. The authors claim the highest efficiency of AQY (full spectrum) among all reports. Although the results are encouraging and characterisation are decent, this work raises concerns in terms of novelty, innovation as well as some fundamental aspects of the observed results. The reviewer considers this work not at the expected high level for Nature Communications. Selective photodeposition of dual cocatalysts on faceted BiVO₄ or other Bi-based ternary oxides has been widely reported. Selective deposition of CoO_x on the oxidative facet {110} and selective deposition of precious metals on reductive facet of {010} have been investigated thoroughly. Following this spatial location of cocatalyst, it performed better than randomly decorated cocatalysts in various redox reactions. In this regard, the novelty and impact are not as high.

Response We thank the reviewer for the opportunity to clarify the novelty of this work. We would like to emphasize that the major novelty of this work is to develop a robust inorganic particulate photocatalyst for efficient photosynthesis of H₂O₂, while developing a new route of photocatalyst fabrication is not the major goal of this study. Improving the efficiency and robustness of photocatalysts are critical for photocatalytic H₂O₂ generation. Existing studies achieving high efficiencies for H₂O₂ generation exclusively rely on organic polymers, yet organic polymers are susceptible to potent oxidants (e.g. •OH) which are inevitably formed during H₂O₂ generation. This issue is highlighted in a very recently paper (<https://pubs.acs.org/doi/full/10.1021/jacs.1c09979>). In contrast, inorganic semiconductors are resistant to •OH-mediated oxidation, yet existing inorganic semiconductors remain inefficient for photocatalytic H₂O₂ generation, as summarized in Table S1:

Table S1. Comparison of photocatalytic H₂O₂ production.

Photocatalyst	Experimental Conditions			Rate ($\mu\text{M h}^{-1}$)	AQY % (420 nm)	AQY % (full spectrum)	STH %	Ref.
	Sacrificial agent	Gas	Light					
Inorganic Photocatalyst								
CoO _x /Mo:BiVO ₄ / Pd	No	O ₂	AM 1.5	1425	5.8	1.2	0.29	This wor k

Pd/TiO ₂	No	Air	AM 1.5	150	-	-	-	2
rGO/TiO ₂ /CoPi	No	O ₂	$\lambda > 320$ nm	60	-	-	-	3
GO	No	Air	Simulated sunlight	50	-	-	-	4
rGO/TiO ₂ /P	No	O ₂	$\lambda > 320$ nm	30	-	-	-	3
Au/BiVO ₄	No	O ₂	$\lambda > 420$ nm	12	0.24	-	-	5
BiVO ₄	No	O ₂	$\lambda > 420$ nm	<0.5	-	-	-	5

(2) Chu, C. H. *et al.* Electronic tuning of metal nanoparticles for highly efficient photocatalytic hydrogen peroxide production. *Acs Catal.* **9**, 626-631 (2019). **H₂O₂ photosynthesis performance: 150 μ M/h**

(3) Moon, G. H., Kim, W., Bokare, A. D., Sung, N. E. & Choi, W. Solar production of H₂O₂ on reduced graphene oxide-TiO₂ hybrid photocatalysts consisting of earth-abundant elements only. *Energy Environ. Sci.* **7**, 4023-4028 (2014). **H₂O₂ photosynthesis performance: 60 μ M/h**

(4) Hou, W. C. & Wang, Y. S. Photocatalytic generation of H₂O₂ by graphene oxide in organic electron donor-free condition under sunlight. *Acs Sustain. Chem. Eng.* **5**, 2994-3001 (2017). **H₂O₂ photosynthesis performance: 50 μ M/h**

(5) Hirakawa, H. *et al.* Au nanoparticles supported on BiVO₄: effective inorganic photocatalysts for H₂O₂ production from water and O₂ under visible light. *Acs Catal.* **6**, 4976-4982 (2016). **H₂O₂ photosynthesis performance: 12 μ M/h**

Our study reports for the first time an inorganic photocatalyst that is resistant to •OH-mediated oxidation and shows a comparable efficiency with organic ones in photocatalytic H₂O₂ generation. The achieved apparent quantum yield of 1.2% and a solar-to-chemical conversion efficiency of 0.29% at full spectrum leverages the record of inorganic photocatalysts by one order of magnitude. Our work demonstrates the feasibility of developing robust inorganic particulate photocatalysts for efficient photocatalytic H₂O₂ generation, therefore presenting a key step towards achieving scalable and cost-effective solar fuel productions.

The achieved 0.29% STH efficiency using BiVO₄ is a big progress in the field of photocatalytic H₂O₂ generation. To provide a comparison, photocatalytic water splitting for H₂ generation has been developed for over 50 years with thousands of papers published, yet only very few systems can reliably split water with solar-to-chemical conversion efficiency over 0.2% (the UV-light-responsive Al-doped SrTiO₃ photocatalysts (*Nature*, 2020, 581, 411–414; *Nature*, 2021, 598, 304–307) and a Z-scheme system consisting of Rh-doped SrTiO₃ and BiVO₄ (*Nat. Mater.*, 2016, 15, 611–615).

Actually, the high novelty and impact of this work have been acknowledged by other four

reviewers. Reviewer 2, 3, 4, and 5 all pointed out that the high solar-to-chemical conversion efficiency achieved by inorganic semiconductor is impressive, for instance:

Reviewer 2 commented: ‘The reported reaction efficiency is impressive (apparent quantum yield = 1.2%, solar-to-chemical conversion efficiency at full spectrum = 0.29%). Then, I recommend publication of this paper in Nature Communications after minor revision.’

Reviewer 3 commented: ‘the solar-to-chemical conversion efficiencies reported in this work outperform by one order of magnitude the values reported so far for inorganic photocatalysts.’

Reviewer 4 commented: ‘achieving an AQY of 1.2% at the full spectrum and a STH of 0.29%, a new record for inorganic semiconductor-based systems.’

Reviewer 5 commented: ‘the authors develop a robust all-inorganic photocatalyst system (CoO_x on {110} / Mo:BiVO₄ / Pd on {010}) for the photocatalytic synthesis of peroxide from water (STC of ~0.29%). Overall, the work is of a very high standard, and I recommend its publication in your journal (minor corrections).’

Further, Reviewer 3 pointed out that even though the photocatalyst fabrication strategy has been utilized before, it is not the main interest of this work and would not jeopardize the novelty of this work: ‘Even though this kind of heterostructures involving faceted BiVO₄ microcrystals have already been widely investigated for water reduction reaction and water oxidation reaction, the main interests of manuscript rely on the use of the Mo as dopant to improve the conductivity of BiVO₄ microcrystals and on the use of many complementary techniques to determine the origin of the improvements evidenced with these heterostructures. Using a suitable combination of steady-state and transient absorption spectroscopy techniques, the authors were able to provide convincing features about the electronic processes involved in the photocatalytic process. The work follows the highest standard in the field of photocatalysis and the conclusions are well-supported by the experimental data.’

Regarding the catalyst preparation method, we have fully acknowledged the pioneer studies on facet-controlled cocatalysts loadings from Can Li’s group (*Nat. Commun.*, 2013, 4, DOI: 10.1038/ncomms2401; *Energ. Environ. Sci.*, 2014, 7, 1369–1376). Those studies have confirmed the significance of charge separation between facets and focused on tuning such charge separation by selectively depositing cocatalysts:

Line 207: ‘These results suggest that selective coloaded of CoO_x and Pd not only improved surface kinetics for O₂ evolution and selectivity for H₂O₂ production as introduced above, but also tuned other critical processes like charge separation.³¹ Furthermore, selectively coloaded dual cocatalysts has been applied to enhance charge separation for sacrificial photocatalytic O₂ evolution.^{21,32,33} To this end, charge separation processes in Mo:BiVO₄, CoO_x/Mo:BiVO₄, Pd/Mo:BiVO₄ and CoO_x/Mo:BiVO₄/Pd were thoroughly studied by transient absorption spectroscopy (TAS).’

Line 445: ‘21. Li, R. G. *et al.* Spatial separation of photogenerated electrons and holes among {010} and {110} crystal facets of BiVO₄. *Nat. Commun.* 4 (2013).’

Line 467: ‘32. Li, R. G., Han, H. X., Zhang, F. X., Wang, D. G. & Li, C. Highly efficient photocatalysts constructed by rational assembly of dual-cocatalysts separately on different facets of BiVO₄. *Energ. Environ. Sci.* 7, 1369-1376 (2014).’

Line 469: ‘33. Qi, Y. *et al.* Redox based visible-light-driven Z-scheme overall water splitting

with apparent quantum efficiency exceeding 10%. *Joule* **2**, 2393-2402 (2018).’

These studies only investigated a half reaction for O₂ evolution in the presence of electron acceptor (e.g. Ag⁺) or in a Z-scheme system. In stark contrast to such a half reaction, our study specifically focuses on a full reaction without any sacrificial reagent. We would like to note that in the field of artificial photosynthesis by particulate photocatalysts, achieving a full reaction is much more challenging than achieving a half reaction, especially when the photocatalyst has a relative narrow bandgap like BiVO₄ (More details was discussed in “Section 2.2 Photocatalytic H₂ or O₂ evolution in sacrificial systems”, “Heterogeneous photocatalyst materials for water splitting”, *Chem. Soc. Rev.*, 2009, 38, 253–278).

Comment 1.2 The authors elaborated that the transformation of CoO_x to CoPi within 15 mins of photocatalytic reaction. Although the CoPi is cocatalysing oxygen evolution, the reviewer believes this “deactivation” is significant.

Response We thank the reviewer for bringing up this important point and opportunity to improve the clarity. We would like to note the difference between the stability of photocatalyst and H₂O₂ production. The photocatalyst CoO_x/Mo:BiVO₄/Pd itself is actually very stable and no deactivation was observed in pure water over repetitive use:

Figure S18b. Time courses of photocatalytic H₂O₂ generation over CoO_x/Mo:BiVO₄/Pd in pure water.

We further tested the performance of CoO_x/Mo:BiVO₄/Pd in seawater which is a desirable solution condition for artificial photosynthesis (*J. Phys. Chem. Lett.*, 2010, 1, 2655–2661). No deactivation was observed over five-round repetitive use:

Figure S21. Time courses of photocatalytic H_2O_2 generation over $\text{CoO}_x/\text{Mo}:\text{BiVO}_4/\text{Pd}$ in sea water. The seawater is prepared by dissolving the sea salt consisted with NaCl (77.6 wt%), KCl (2.1 wt%), MgCl_2 (6.6 wt%), CaCl_2 (3.3 wt%), MgSO_4 (9.6 wt%), NaHCO_3 (0.6 wt%), NaBr (0.16 wt%), Na_2SiO_3 (0.007 wt%), NaSi_4O_9 (0.006 wt%), H_3PO_4 (0.006 wt%), H_3BO_3 (0.2 wt%), LiNO_3 (0.004 wt%), Al_2Cl_6 (0.04 wt%) in the DI water.

To provide further evidence for photocatalyst stability, we compared the XPS spectrum before and after $\cdot\text{OH}$ -mediated oxidation. The result shows no difference on Bi, V and O XPS spectra, confirming the high stability of $\text{CoO}_x/\text{Mo}:\text{BiVO}_4/\text{Pd}$. We have added these results and related discussions in the revised manuscript and SI:

Figure S17. The Bi, V and O XPS spectra of $\text{CoO}_x/\text{Mo}:\text{BiVO}_4/\text{Pd}$ before and after $\cdot\text{OH}$ aging for 24h.

Line 152: ‘Most importantly, in stark contrast to the high $\cdot\text{OH}$ -susceptibility of organic photocatalyst, after 24-h incubation under $\cdot\text{OH}$ -rich conditions, $\text{CoO}_x/\text{Mo}:\text{BiVO}_4/\text{Pd}$ exhibits nominal change in H_2O_2 productions (Fig. S1b) and chemical compositions (Fig. S17), demonstrating its high resistance to $\cdot\text{OH}$ -mediated oxidation.’

Moreover, as shown in Fig. S18b and also observed in many other H_2O_2 photosynthesis studies, the cumulative H_2O_2 production over time is not linear, which is caused by H_2O_2 decomposition rather than the deactivation of photocatalysts. Therefore, PO_4^{3-} was applied to inhibit the decomposition of H_2O_2 , and to improve the accuracy of activity tests. The addition of PO_4^{3-} is quite common in H_2O_2 production performance tests, for instance:

Shiraishi, Y., Ueda, Y., Soramoto, A., Hinokuma, S. & Hirai, T. Photocatalytic hydrogen peroxide splitting on metal-free powders assisted by phosphoric acid as a stabilizer. *Nat.*

Commun. **11** (2020).

Greenspan, F. P. Method of improving stability of concentrated hydrogen peroxide in contact with stainless and aluminum alloys. US patent US2782100A (1957).

Shimokawa, S., Namikawa, K. & Murakami, S. Stabilized refined aqueous hydrogen peroxide solution. Jpn. Kokai Tokkyo Koho JP07-081906A (1995).

For large-scale applications of a particulate photocatalyst (also a potential research direction for applying our $\text{CoO}_x/\text{Mo}:\text{BiVO}_4/\text{Pd}$ photocatalyst), the photocatalyst is typically immobilized on a support using drop-casting (*Energy Technology*, 2015, **3**, 1014–1017) or screen printing (*Nat. Mater.*, 2016, **15**, 611–615) technologies. Then, the reaction will be conducted with flow liquid. In those setups, the photogenerated H_2O_2 will be immediately removed from catalyst surface to prevent H_2O_2 decomposition. Naturally, the addition of PO_4^{3-} and resultant transformation of CoO_x to CoPi will be avoided. For instance, in our previous studies, we fabricated a 1×1 m $\text{Al}:\text{SrTiO}_3$ panel by drop-casting as a prototype for large-scale solar hydrogen generation (see movie: <https://ars.els-cdn.com/content/image/1-s2.0-S2542435117302246-mmc3.mp4>; *Joule*, 2018, **2**, 509–520). By immobilizing the photocatalyst, packing of photocatalyst was prohibited and photocatalyst showed stable activity for at least 7h under the intense illumination as an accelerated deactivation test (*Joule*, 2018, **2**, 509–520). In our recent work, the feasibility of applying immobilized photocatalyst with high solar-to-chemical energy conversion efficiency was demonstrated on a 100-m^2 scale (*Nature*, 2021, **598**, 304–307).

Referring to our previous large-scale photolysis device setup, we plan to fabricate the $\text{CoO}_x/\text{Mo}:\text{BiVO}_4/\text{Pd}$ panel where SiO_2 nanoparticles (diameter ca. 20 nm) will be used as an inorganic binder to firmly fix a $\text{CoO}_x/\text{Mo}:\text{BiVO}_4/\text{Pd}$ layer onto the substrate (e.g., glass, *Joule*, 2018, **2**, 509–520). The device validation and performance test are being conducted in ongoing works. As a first step, we designed a flow cell photolysis system, where photogenerated H_2O_2 is immediately separated from $\text{CoO}_x/\text{Mo}:\text{BiVO}_4/\text{Pd}$ particles to avoid H_2O_2 decomposition, and therefore no PO_4^{3-} addition is required.

Figure R1. Photograph and schematic illustration of the flow cell device for photocatalytic H_2O_2 productions.

We have added related discussions in the revised manuscript to clarify the stability of $\text{CoO}_x/\text{Mo}:\text{BiVO}_4/\text{Pd}$:

Line 185: ‘As expected, $\text{CoO}_x/\text{Mo}:\text{BiVO}_4/\text{Pd}$ was highly stable over five cycles of reaction. To further demonstrate the stability of $\text{CoO}_x/\text{Mo}:\text{BiVO}_4/\text{Pd}$, we test the performance of $\text{CoO}_x/\text{Mo}:\text{BiVO}_4/\text{Pd}$ in seawater which is a desirable solution condition for artificial photosynthesis.²⁸ No deactivation was observed over five-round repetitive use (Fig. S21). Yet the cumulative production of H_2O_2 was lower than that in phosphate solution owing to H_2O_2 decomposition, consistent with the previous report.²⁹’

Line 188: ‘Yet the cumulative production of H_2O_2 was lower than that in phosphate solution owing to H_2O_2 decomposition, consistent with the previous report.²⁹ In following studies, the cumulative production of H_2O_2 in pure water will be improved with rapid H_2O_2 diffusion by a large-scale photosynthesis setup where the $\text{CoO}_x/\text{Mo}:\text{BiVO}_4/\text{Pd}$ photocatalyst will be immobilized on a support using drop-casting or screen printing technologies and integrated in a flow cell photolysis system.^{19,30}’

Line 463: ‘30. Schroder, M. *et al.* Hydrogen evolution reaction in a large-scale reactor using a carbon nitride photocatalyst under natural sunlight irradiation. *Energy Technol-Ger* **3**, 1014-1017 (2015).’

Comment 1.3 *The thus-formed H_2O_2 can go through the further oxidative decomposition by BiVO_4 . Oxidation of H_2O_2 and water would become competitive reactions. This might also be the reason of non-linear observation of H_2O_2 with time. The authors may want to look into this aspect. If H_2O_2 is not separated from the photocatalytic system upon generation, basically the system may not last long.*

Response This question is similar to Comment 1.2. We have performed additional experiments and analyses and revised the manuscript thanks to this question. Since our response is extensive, we consolidated our answers above.

Comment 1.4 *The authors stated that the reaction is determined by the charge-separation efficiency. However, the reviewer think that it is the charge transfer mechanism that determines the selectivity of ORR to H_2O_2 instead of other radicals. Charges are separated within the photocatalyst but the charge transfer affected the product selectivity.*

Response We agree with the reviewer that H_2O_2 production selectivity is an important factor for photocatalytic H_2O_2 generation. For this reason, we loaded Pd on $\text{Mo}:\text{BiVO}_4$ to enhanced the selectivity of H_2O_2 generation as shown in Fig. S15.

Figure S15. Selectivity of H₂O₂ production. Reaction conditions: photocatalyst amount, 2 mg; reactant solution, 12 mL PBS solution (pH=7.4) saturated with O₂; light source, xenon lamp solar simulator, 100 mW/cm², AM 1.5G.

Yet, we believe that charge separation is another important factor for efficient H₂O₂ production. Even though the selectivity of 83% for H₂O₂ generation on CoO_x/Mo:BiVO₄/Pd is already high, the achieved apparent quantum yield at full spectrum is still only 1.2%, indicating that the majority of charge carriers recombined and charge separation is a major factor determining the H₂O₂ production efficiency. The importance of the charge-separation processes for photocatalytic reactions has been well acknowledged in the previous studies (*ChemSusChem*, 2017, 10, 4277–4305). Charges are separated in the bulk of a photocatalyst as the reviewer indicated, but such processes are driven by the photocatalyst's surface energetics which is determined by the optoelectronics of the photocatalysts and the loaded cocatalysts (*Sustainable Energy Fuels*, 2019, 3, 850–864; *Energy Environ., Sci.*, 2020, 13, 162–173). To this end, a cocatalyst is applied to improve not only the surface kinetics and selectivity, but also the surface energetics for charge separation (*Acc. Chem. Res.*, 2013, 46, 8, 1900-1909). For our system, after loading CoO_x cocatalyst, the H₂O₂ production of CoO_x/Mo:BiVO₄/Pd is 6-fold higher than that of Mo:BiVO₄/Pd. More importantly, the enhancement on H₂O₂ production by coloaded Pd and CoO_x is even higher than the multiplication of individual enhancements. In the meantime, when CoO_x and Pd were randomly deposited on Mo:BiVO₄, it was only 32% as active as CoO_x/Mo:BiVO₄/Pd, though these two catalysts had similar surface kinetics and selectivity. These results suggest that selective coloaded of CoO_x and Pd not only improved surface kinetics and selectivity as introduced above, but also tuned charge separation.

Moreover, we conducted very thorough transient absorption spectroscopy to study the charge-separation processes. TAS results clearly demonstrate that selectively coloaded Pd and CoO_x on the expected facets of Mo:BiVO₄ significantly enhanced the charge separation and suppressed rapid charge-carrier trapping and recombination. The suppressed charge-carrier trapping and recombination, a key challenge in improving the efficiency of inorganic photocatalysts, are critical for achieving efficient H₂O₂ photosynthesis.

We have revised the manuscript to further clarify the enhanced charge separation by

selective coloaded of CoO_x and Pd:

Line 207: ‘These results suggest that selective coloaded of CoO_x and Pd not only improved surface kinetics for O₂ evolution and selectivity for H₂O₂ production as introduced above, but also tuned other critical processes like charge separation.’³¹

Comment 1.5 Grammatical errors can be spotted throughout the manuscript: e.g.

- The TA signals ...was still

- To clarify such decay ...:

Response We thank the reviewer for the opportunity to improve the clarity. We have revised the grammatical errors that were pointed out. We further revised the whole manuscript accordingly. The revision involving grammatical errors were highlighted in purple.

Line 170: ‘To clarify such decay, CoO_x/Mo:BiVO₄/Pd was tested with cycles of reaction.’

Line 233: ‘Regardless of such an unfavorable effect, the TA signals for this kind of photocarriers were still observed to vary significantly after loading Pd or CoO_x on Mo:BiVO₄ (Fig. 3b).’

Line 13: ‘This set of photocatalysts is susceptible to potent oxidants (e.g. hydroxyl radical) that are inevitably formed during H₂O₂ generation.’

Line 23: ‘The promising H₂O₂ generation efficiency achieved by delicate design of catalyst spatial and electronic structures sheds light on applying robust inorganic particulate photocatalysts to artificial photosynthesis of H₂O₂.’

Line 31: ‘Among primary photosynthetic systems, including photovoltaic-assisted electrolysis,⁷ photoelectrochemical catalysis,^{8,9} and particulate photocatalysis (PC),¹⁰ PC is the most cost-effective because of its simplicity and scalability.’¹¹

Comment 1.6 How would the deep trapped electrons be able to move to Pd as illustrated in Figure 4c?

Response We thank the reviewer for the opportunity to improve the clarity. Transfer of electrons trapped at deep states to Pd cocatalyst is possible when these trapped electrons in BVO₄:Mo diffuse via tunneling and hopping from trap to trap. In this manner, the transfer of electrons trapped at the defects in the vicinity of Pd particles occurs. This electron transfer involving deeply trapped electrons have been observed also in several photocatalysts such as RhCr_xO_y/SrTiO₃:Na (*ChemCatChem*, 2019, 11, 6349–6354) and (Fe,Ru)O_x/ Bi₄TaO₈Cl (*ACS Appl. Mater. Interfaces*, 2019, 11, 49, 45606–45611). A similar diffusion process involving deeply trapped electrons (redistribution of trapped electrons from deep to shallow traps) is often proposed to take place on long-persistence phosphors materials (*Inorg. Chem.*, 2018, 57, 9, 5194–5203). Similar process can also happen on photocatalysts containing deep defect states. In fact, in the field of developing long-lived phosphors, the deep electron trapping is not a rare case, wherein the trap states are situated 0.5~1 eV below the conduction band minimum (CBM). Aside from this diffusion process, the re-excitation of deeply trapped electrons to the CB will also be possible.

We have revised the manuscript to further clarify that the deep trapped electrons are able

to move to Pd:

Line 270: ‘This result suggests that the deeply trapped electrons transferred to Pd in Mo:BiVO₄:Mo and CoO_x/Mo:BiVO₄ /Pd and became available for subsequent surface reactions. The charge transfer involving deeply trapped electrons to Pd cocatalyst is possible via tunneling and trap-to-trap hopping. A similar diffusion of trapped electrons is often proposed to take place on long-lived persistent phosphor materials,³⁷ wherein the trap states are situated at ~0.5-1.0 eV below the conduction band minimum. Aside from this diffusion process, re-excitation of deeply trapped electrons to the CB and eventually transfer to the Pd will be possible. Taken together, the above findings demonstrate that selectively coloaded Pd and CoO_x on the expected facets of Mo:BiVO₄ significantly enhanced the charge separation and suppressed rapid charge-carrier trapping and recombination.’

Line 478: ‘37. Xu, J., Murata, D., Ueda, J., Viana, B. & Tanabe, S. Toward rechargeable persistent luminescence for the first and third biological windows via persistent energy transfer and electron trap redistribution. *Inorg. Chem.* **57**, 5194-5203 (2018).’

Comment 1.7 Efficiencies (AQY and other) are higher but the reaction time of 15 mins are relatively not practical.

Response We note that AQY and STH were tested with a reaction time of 1 hour, as indicated in the manuscript:

$$AQY = \frac{2 \times [H_2O_2]_{1h} \times V}{I_{tot,p} \times A \times 3600} \quad STH = \frac{\Delta G(H_2O_2) \times [H_2O_2]_{1h} \times V}{I_{tot,e} \times A \times 3600}$$

We have also revised the manuscript and SI to clarify the reaction time:

$$AQY = \frac{2 \times [H_2O_2]_{1h} \times V}{I_{tot,p} \times A \times t}$$

Line 340: ‘where [H₂O₂]_{1h} is the H₂O₂ concentration, I_{tot,p} is the total photo flux of simulated sunlight irradiation (4.4 × 10⁻³ mol/m²/s, calculation details were shown in Section S3), V is the volume of suspension (12 mL), A is the irradiation area (1.91 cm² in this study) and t is the reaction time (1h).’

$$STH = \frac{\Delta G(H_2O_2) \times [H_2O_2]_{1h} \times V}{I_{tot,e} \times A \times t}$$

Line 345: ‘Where [H₂O₂]_{1h} is the free energy for H₂O₂ formation (117 kJ mol⁻¹), I_{tot,e} is the total intensity of simulated sunlight irradiation (0.1 W/cm²), V is the volume of suspension (12 mL), A is the irradiation area (1.91 cm² in this study) and t is the reaction time (1h).’

Reviewer 2

Comment 2.1 *First of all, I apologize the delay in reviewing this paper. The authors report photocatalytic synthesis of hydrogen peroxide from water and oxygen using a Mo:BiVO₄ photocatalyst with the {110} and {010} facets modified by CoO_x and Pd, respectively. The reported reaction efficiency is impressive (apparent quantum yield = 1.2%, solar-to-chemical conversion efficiency at full spectrum = 0.29%). Then, I recommend publication of this paper in Nature Communications after minor revision.*

Response We thank the reviewer for the time and effort to review our manuscript as well as the encouraging comments.

Comment 2.2 *Lines 131, 132, 184, 185: The enhancement factors should be expressed by taking the significant figures of the data in Figure 2a.*

Response We thank the reviewer for the opportunity to improve the clarity. We have revised the manuscript accordingly:

Line 135: 'Bare Mo:BiVO₄ exhibits minimal performance for 60-min H₂O₂ generation (4.1 μM). Loading CoO_x cocatalyst enhanced the photocatalytic H₂O₂ generation performance by a factor of 1.8 (7.5 μM, Fig. 2a), attributing to promoted water oxidation and consequentially reduced detrimental charge recombination (Fig. 2c). In the meanwhile, loading Pd cocatalyst onto Mo:BiVO₄ improved H₂O₂ generation selectivity (Fig. 2c), resulting in a 53.7-fold enhancement of 60-min H₂O₂ generation (220.0 μM, Fig. 2a).'

Comment 2.3 *Lines 215-218: This part should be explained more clearly and in more detail.*

Response We thank the reviewer for the opportunity to improve the clarity. We have added more details to the manuscript accordingly:

Line 236: 'In contrast, loading CoO_x on the {110} facet lead to an opposite effect because CoO_x captured holes and thus increased the electron population in Mo:BiVO₄. At 50 ps, for instance, loading CoO_x increased the intensity of TA signal for free/shallowly trapped electrons by ~66%, while loading Pd only decrease the intensity by ~6% compared to that of bare sample. The effect of CoO_x on the dynamics of the free/shallowly trapped electrons was more intense than that of Pd because electron transfer to Pd competed with trapping of electrons into deep trap states below the CB. Electron trapping to deep trap states was represented by the strong TA signal probed at 781 nm as shown in Fig. 3a.'

Comment 2.4 *Lines 225-229: These sentences should be rewritten.*

Response We thank the reviewer for the opportunity to improve the clarity. We have rewritten the sentences in the manuscript accordingly:

Line 248: 'This result indicates that in this time region, Pd and CoO_x captured electrons and holes, respectively, as expected. The population of free/shallowly trapped electrons that remained in Mo:BiVO₄ after electrons are captured by Pd can be determined by estimating the ratio of ΔAbsorbance displayed in Fig. 4a with respect to bare Mo:BiVO₄ and

CoO_x/Mo:BiVO₄ for Mo:BiVO₄/Pd and CoO_x/Mo:BiVO₄/Pd, respectively. At 200 μs, for instance, 59% and 42% of free/shallowly trapped electrons remained in Mo:BiVO₄ for Mo:BiVO₄/Pd and CoO_x/Mo:BiVO₄/Pd, respectively. This implies that 41% and 58% of free/shallowly trapped electrons transferred to Pd for Mo:BiVO₄/Pd and CoO_x/Mo:BiVO₄/Pd, respectively. More electrons transferred to Pd on CoO_x/Mo:BiVO₄/Pd than on Mo:BiVO₄/Pd because of more efficient charge separation (synergistic charge separation) in the former case. To further justify this effect, the impact of CoO_x and Pd on the decay kinetics of trapped holes in Mo:BiVO₄ was investigated.'

Comment 2.5 Lines 238-240: The data for CoO_x/BiVO₄:Mo/Pd is missing in Fig. S24.

Response We have added the data for CoO_x/Mo:BiVO₄/Pd in Fig.S24:

Figure S24. Transient profiles of photocarriers probed at 781 nm (deeply trapped electrons) for Mo:BiVO₄, Mo:BiVO₄/Pd, CoO_x/Mo:BiVO₄ and CoO_x/Mo:BiVO₄/Pd.

Comment 2.6 Lines 240-242: The sentence should be corrected.

Response We thank the reviewer for the opportunity to improve the clarity. Transfer of electrons trapped at deep states to Pd cocatalyst is possible when these trapped electrons in BVO₄:Mo diffuse via tunneling and hopping from trap to trap. In this manner, the transfer of electrons trapped at the defects in the vicinity of Pd particles occurs. This electron transfer involving deeply trapped electrons have been observed also in several photocatalysts such as RhCr_xO_y/SrTiO₃:Na (*ChemCatChem*, 2019, 11, 6349–6354) and (Fe,Ru)O_x/Bi₄TaO₈Cl (*ACS Appl. Mater. Interfaces*, 2019, 11, 49, 45606–45611). A similar diffusion process involving deeply trapped electrons (redistribution of trapped electrons from deep to shallow traps) is often proposed to take place on long-persistence phosphors materials (*Inorg. Chem.*, 2018, 57, 9, 5194–5203). Similar process can also happen on photocatalysts containing deep defect states. In fact, in the field of developing long-lived phosphors, the deep electron trapping is not a rare case, wherein the trap states are situated 0.5~1 eV below the conduction band minimum (CBM). Aside from this diffusion process, the re-excitation of deeply trapped electrons to the CB will also be possible.

We have revised the manuscript to further clarify that the deep trapped electrons are able to move to Pd accordingly to the manuscript :

Line 270: 'This result suggests that the deeply trapped electrons transferred to Pd in Mo:BiVO₄/Pd and CoO_x/Mo:BiVO₄/Pd and became **available** for subsequent surface

reactions. The charge transfer involving deeply trapped electrons to Pd cocatalyst is possible via tunneling and trap-to-trap hopping. A similar diffusion of trapped electrons is often proposed to take place on long-lived persistent phosphor materials,³⁷ wherein the trap states are situated at ~0.5-1.0 eV below the conduction band minimum. Aside from this diffusion process, re-excitation of deeply trapped electrons to the CB and eventually transfer to the Pd will be possible. Taken together, the above findings demonstrate that selectively coloaded Pd and CoO_x on the expected facets of Mo:BiVO₄ significantly enhanced the charge separation and suppressed rapid charge-carrier trapping and recombination.'

Line 478: '37. Xu, J., Murata, D., Ueda, J., Viana, B. & Tanabe, S. Toward rechargeable persistent luminescence for the first and third biological windows via persistent energy transfer and electron trap redistribution. *Inorg. Chem.* **57**, 5194-5203 (2018).'

Comment 2.7 *Fig. 4: I would like to know the origin for the band bending between the {110} and {010} facets. Also, does it exist not only on the surface of BiVO₄ but also in the bulk?*

Response We thank the reviewer for raising this important discussion point. The facet dependent energetics was first confirmed on anatase TiO₂ and such phenomenon was caused by the variation in surface termination atoms (*Nature* 2008, 453, 638–641). While the origin of the band bending between {110} and {010} facets on BiVO₄ has not been investigated, we believe that it will be similar with that for faceted TiO₂.

The band bending exists on both the surface and the bulk of BiVO₄. The band bending extends from surface to bulk, constructing a space-charge region. Such band bending phenomenon has been nicely imaged by a recent work from Can Li's group using spatially resolved surface photovoltage spectroscopy (see below) (*Nat. Commun.*, 2013, 4, DOI: 10.1038/ncomms2401; *Angew. Chem. Int. Edit.*, 2015, 54, 9111–9114; *Nano Lett.*, 2017, 17, 6735–6741). We would like to emphasize that the spatial charge separation of a single crystal is attributed to the band bending of different facets, and loading a cocatalyst can tune the surface energetics for enhancing charge separation.

Figure R2. Schematic of the built-in electric field with relative strength in the SCR of different facets (*Angew. Chem. Int. Edit.*, 2015, **54**, 9111-9114).

To further illustrate the band bending between different facets, we have added a new scheme in the revised SI and revised the manuscript:

Figure S25. Schematic band diagrams across the border between the {011} and {010} facets of a bare single Mo:BiVO₄ photocatalyst particle and facets of a single Mo:BiVO₄ photocatalyst particle with CoO_x cocatalyst selectively deposited on {010} facet (green line) and Pd cocatalyst selectively deposited on {010} facet (orange line), respectively. The facet dependent energetics on a bare single Mo:BiVO₄ photocatalyst particle was caused by the variation in surface termination atoms.¹⁶

Line 280: ‘Such positive effects are supposed to be achieved by tuning the energetics between cocatalysts and respective Mo:BiVO₄ facet (Figs. 4c and S25).’

Reviewer 3

Comment 3.1 *This manuscript reports on a very nice piece of works dealing with the use of CoO_x/Mo:BiVO₄/Pd heteromicrostructures for the efficient production of H₂O₂ by photocatalysis under solar light. First of all, the solar-to-chemical conversion efficiencies reported in this work outperform by one order of magnitude the values reported so far for inorganic photocatalysts. Even though this kind of heterostructures involving faceted BiVO₄ microcrystals have already been widely investigated for water reduction reaction and water oxidation reaction, the main interests of manuscript rely on the use of the Mo as dopant to improve the conductivity of BiVO₄ microcrystals and on the use of many complementary techniques to determine the origin of the improvements evidenced with these heterostructures. Using a suitable combination of steady-state and transient absorption spectroscopy techniques, the authors were able to provide convincing features about the electronic processes involved in the photocatalytic process. The work follows the highest standard in the field of photocatalysis and the conclusions are well-supported by the experimental data.*

Response We thank the reviewer for the time and effort to review our manuscript as well as the encouraging comments. We are humbled by the reviewer's praises.

Comment 3.2 *The controlled photodeposition of metallic particles or metal oxide particles on BiVO₄ microparticles exposing well-defined facets in order to obtain heterostructures has already been reported by different groups. The authors should therefore quote some references about this point, especially the pioneer's work of Can Li et al. (Nat. Commun., 2013, 4, 1432 and Ener. Environ. Sci., 2014, 7, 1369) or more recent ones for efficient water oxidation reactions.*

Response We thank the reviewer for the suggestion on acknowledging those pioneer works. We have added more citations and revised the manuscript accordingly:

Line 207: 'These results suggest that selective coloaded of CoO_x and Pd not only improved surface kinetics for O₂ evolution and selectivity for H₂O₂ production as introduced above, but also tuned other critical processes like charge separation.³¹ Furthermore, selectively coloaded dual cocatalysts has been applied to enhance charge separation for sacrificial photocatalytic O₂ evolution.^{21,32,33} To this end, charge separation processes in Mo:BiVO₄, CoO_x/Mo:BiVO₄, Pd/Mo:BiVO₄ and CoO_x/Mo:BiVO₄/Pd were thoroughly studied by transient absorption spectroscopy (TAS).'

Line 445: '21. Li, R. G. et al. Spatial separation of photogenerated electrons and holes among {010} and {110} crystal facets of BiVO₄. Nat. Commun. 4 (2013).'

Line 467: '32. Li, R. G., Han, H. X., Zhang, F. X., Wang, D. G. & Li, C. Highly efficient photocatalysts constructed by rational assembly of dual-cocatalysts separately on different facets of BiVO₄. Energ. Environ. Sci. 7, 1369-1376 (2014).'

Line 469: '33. Qi, Y. et al. Redox based visible-light-driven Z-scheme overall water splitting with apparent quantum efficiency exceeding 10%. Joule 2, 2393-2402 (2018).'

Comment 3.3 *A scheme or photograph of the experimental set-up should be shown in the SI file in order to be able to reproduce the experiments. Moreover, the emission spectrum of the Xenon*

lamp employed should be provided.

Response

We thank the reviewer for the opportunity to improve the clarity. We have provided more information on the experimental set-up:

Figure S28. Photographs (a) and schematic illustration (b) of the devices used in the photocatalysis performance tests.

We also added the spectrum of the Xenon lamp and more information in the revised SI and manuscript:

Figure S29. Spectrum of the Xenon lamp (orange line) and the standard AM 1.5G (blue line, ASTM G 173).

Line 324: ‘The light intensity was adjusted to 100 mW/cm² (AM 1.5G; irradiation area = 1.83 cm²). The Xenon lamp and the standard AM1.5G (ASTMG 173) spectrum is shown in Figure S29. For the wavelength-dependent AQY analysis, the photolysis was performed using LED light irradiation (model slight; Perfect Light, Inc.).’

Comment 3.4 The actual chemical composition of the heterostructures should be determined more precisely. First of all, the actual presence of Mo was not evidenced. Moreover, the experimental bulk Pd/Bi and Co/Bi atomic ratios should be reported because it seems that the amount reported in the article are nominal loadings. XPS and EDX analyses would provide information about the surface and bulk composition of the materials prepared.

That would improve the manuscript.

Response

We thank the reviewer's comment for clarifying the elemental composition. Following the reviewer's suggestions, we conducted more XPS and EDX element analyses on $\text{CoO}_x/\text{Mo}:\text{BiVO}_4/\text{Pd}$ surface. The presence of Mo in $\text{Mo}:\text{BiVO}_4$ was confirmed by XPS as shown in Fig. S26. Because the doping amount of Mo was minor, XPS or EDX was not able to accurately confirm the amount of Mo. Therefore, we conducted further ICP-MS analysis after sample digestion. The result shows that atomic Mo/V doping amount was 0.023%.

Further, XPS results show that the Pd/Bi and Co/Bi atomic ratios on $\text{CoO}_x/\text{Mo}:\text{BiVO}_4/\text{Pd}$ surface are 2.0% and 1.7%, respectively. The EDX results show that the Pd/Bi and Co/Bi atomic ratios are 2.3% and 2.2%. ICP-MS results show that the bulk Pd/Bi and Co/Bi were 0.34% and 0.22%, respectively.

We have summarized these results in Table S3 and added corresponding data in the revised manuscript and SI:

Figure S26. The full XPS spectrum of $\text{CoO}_x/\text{Mo}:\text{BiVO}_4/\text{Pd}$.

Figure S27. The EDX spectrum of $\text{CoO}_x/\text{Mo}:\text{BiVO}_4/\text{Pd}$.

Table S3. The Mo/V, Pd/Bi and Co/Bi atomic yield of $\text{CoO}_x/\text{Mo}:\text{BiVO}_4/\text{Pd}$

	Mo/V	Pd/Bi	Co/Bi
XPS	-	2.0%	1.7%
EDS	-	2.3%	2.2%
ICP	0.023%	0.34%	0.22%

Line 305: ‘ $\text{Mo}:\text{BiVO}_4\text{-CoO}_x\text{-Pd}$ was prepared following an impregnation procedure by air-purging the mixture of $\text{Mo}:\text{BiVO}_4$ (0.2 g), $\text{Co}(\text{NO}_3)_2$ (0.4 mg) and Na_2PdCl_4 (0.8 mg) (from stock) until dry, followed by heating in a ceramic crucible at a heating rate of $5\text{ }^\circ\text{C}/\text{min}$ to $200\text{ }^\circ\text{C}$ and annealing for 0.5 h under reductive condition (10% H_2 and 90% Ar) in a tube furnace. The elemental compositions were analyzed by EDX, XPS, and ICP-MS (Figs. S26-27 and Table S3).’

Line 316: ‘UV-DRS spectra was taken with a Shimadzu UV-3600 with a resolution of 0.1 nm. The ICP-MS measurement was performed with a NexION 300X (detection limit $1\text{ }\mu\text{g}/\text{L}$).’

Comment 3.5 *Photocatalytic properties of a material are related to surface properties. As a result the question arises about the evolution of the specific areas when the co-catalysts are deposited. The reviewer would therefore suggest to give the specific surface areas of the different photocatalysts studied and to comment these values. Moreover, what is the influence of the sizes of the Pd and CoO_x nanoparticles on the photocatalytic properties? Are the sizes of these nanoparticles reproducible from one batch to another? If not, does it influence the photocatalytic properties?*

Response We thank the reviewer for raising this interesting discussion point. Following the reviewer’s suggestion, we assessed the surface areas of different photocatalysts ($\text{Mo}:\text{BiVO}_4$, $\text{CoO}_x/\text{Mo}:\text{BiVO}_4$, $\text{Mo}:\text{BiVO}_4/\text{Pd}$ and $\text{CoO}_x/\text{Mo}:\text{BiVO}_4/\text{Pd}$) by Brunner–Emmet–Teller (BET) test with the nitrogen as the analysis absorptive at 77.2 K . The results show that the surface areas of four investigated materials are similar

(1.43-1.71 m²/g), as shown in Table S2. The surface areas show the following trend: Mo:BiVO₄ < CoO_x/Mo:BiVO₄ ≈ CoO_x/Mo:BiVO₄/Pd < CoO_x/Mo:BiVO₄/Pd. Since their variation is small, we believe that the surface area has negligible impact when compare the photocatalytic H₂O₂ activities of these samples. We have added the results in the revised manuscript and SI accordingly:

Table S2. The result of BET test.

Sample	Mo:BiVO ₄	CoO _x /Mo:BiVO ₄	Mo:BiVO ₄ /Pd	CoO _x /Mo:BiVO ₄ /Pd
BET (m²/g)	1.43	1.56	1.60	1.71

Line 72: ‘The X-ray diffraction (XRD) pattern of Mo:BiVO₄ as well as CoO_x/Mo:BiVO₄, Mo:BiVO₄/Pd and CoO_x/Mo:BiVO₄/Pd particles matched well with that of monoclinic BiVO₄, with {010} and {110} facet peaks located at 30.6° and 18.7°, respectively (Fig. S3). The Brunner–Emmet–Teller (BET) tests show Mo:BiVO₄ as well as CoO_x/Mo:BiVO₄, Mo:BiVO₄/Pd and CoO_x/Mo:BiVO₄/Pd particles exhibit similar surface areas (1.43-1.71 m²/g, Table S2). The Mo:BiVO₄ particles exhibit a decahedron structure with clear facets as shown in scanning electron microscope (SEM) images (Figs. 1a and 1b).’

Line 312: ‘The XRD patterns were recorded with a Bruker D8 Advance X-ray diffractometer with Cu Kα radiation (λ=1.5406 Å) operated at 40 kV and 40 mA. The BET tests were performed by an ASAP 2460 with N₂ analysis adsorptive at 77.2 K. SEM images were taken using a Hitachi SU-8010 microscope equipped with EDS at 30 kV.’

The sizes of cocatalyst particles will affect the surface energetics at the photocatalyst/cocatalyst interface and thus the charge-separation processes which determines the photocatalytic H₂O₂ generation activity. When the size of cocatalyst particles is smaller than a critical value, the photocatalyst/cocatalyst junction will be pinched-off. In this case, the surface energetics at the photocatalyst/cocatalyst interface will be influenced by electrolyte. When the size of cocatalyst particles is larger than the critical value, the photocatalyst/cocatalyst junction will form be a buried junction. In this case, the surface energetics at the photocatalyst/cocatalyst interface will be solely influenced by the cocatalyst. Such an effect of cocatalyst particle size has been elaborated in photoelectrode systems in previous studies (*J. Phys. Chem., B* 2001, 105, 12303–12318 12303; *Nature Materials*, 2020, 19, 69–76), but it has not been well studied in particulate photocatalysts systems. While we are highly interested in this direction, the analysis on such nanoscale photocatalyst/cocatalyst interfaces requires special devices like a potential-sensing electrochemical atomic force microscope. Such discussions also involve an independent study and are out of the scope of this study.

To check if the sizes of cocatalysts are reproducible, we synthesized three batches of CoO_x (2 wt%) /Mo:BiVO₄/Pd (0.4 wt%) photocatalysts SEM results show that the sizes of Pd and CoO_x nanoparticles are very similar among photocatalysts prepared under different batches (see the figure blow), indicating that the sizes of nanoparticles are reproducible from one batch to another.

Figure R3. Three batches of $\text{CoO}_x(2\%)/\text{Mo:BiVO}_4/\text{Pd}(0.4\%)$ photocatalysts SEM images.

Reviewer 4

Comment 4.1 *In this manuscript, the authors prepared faceted Mo:BiVO₄ particles selectively loaded with CoO_x and Pd as WOR and ORR cocatalysts on {110} and {010} facets for efficient overall photocatalytic H₂O₂ generation, achieving an AOY of 1.2% at the full spectrum and a STH of 0.29%, a new record for inorganic semiconductor-based systems. However, there are some problems needed to be solved before it can be recommended for publication.*

Response We thank the reviewer for the time and effort to review our manuscript as well as the encouraging comments. Thanks to the reviewer's thoughtful comments, we were able to further improve the quality of our manuscript.

Comment 4.2 *The authors only present the XRD pattern of Mo:BiVO₄. After loading cocatalysts onto Mo:BiVO₄, the authors should also provide their XRD patterns.*

Response We thank the reviewer for the opportunity to improve the clarity. Following the reviewer's suggestion, we assessed the XRD pattern of CoO_x/Mo:BiVO₄, Mo:BiVO₄/Pd and CoO_x/Mo:BiVO₄/Pd. The result shows no significant change in XRD patterns before and after cocatalyst loadings, indicating the stable crystal structure of Mo:BiVO₄ during the photo-deposition. We have added the results in SI and revised the manuscript accordingly:

Figure S3. The XRD pattern of Mo:BiVO₄. The XRD patterns of Mo:BiVO₄, CoO_x/Mo:BiVO₄, Mo:BiVO₄/Pd and CoO_x/Mo:BiVO₄/Pd are in good agreement with the JCPDS standard card #14-0688, corresponding to monoclinic scheelite BiVO₄.

Line 72: 'The X-ray diffraction (XRD) pattern of Mo:BiVO₄ as well as CoO_x/Mo:BiVO₄, Mo:BiVO₄/Pd and CoO_x/Mo:BiVO₄/Pd particles matched well with monoclinic BiVO₄, with

{010} and {110} facet peaks located at 30.6° and 18.7° , respectively (Fig. S3). The Brunner–Emmet–Teller (BET) tests show Mo:BiVO₄ as well as CoO_x/Mo:BiVO₄, Mo:BiVO₄/Pd and CoO_x/Mo:BiVO₄/Pd particles exhibit similar surface areas (1.43-1.71 m²/g, Table S2). The Mo:BiVO₄ particles exhibit a decahedron structure with clear facets as shown in scanning electron microscope (SEM) images (FigS. 1a and 1b).⁷

Comment 4.3 More evidence should be provided for the Facet-selective loading of CoO_x and Pd cocatalysts on Mo:BiVO₄, such as high-resolution TEM.

Response We thank the reviewer for suggestions on providing more visualized evidence on selective loadings of cocatalysts. We have tried to analyze these cocatalysts by HR-TEM, unfortunately the Mo:BiVO₄ particle was too thick and the electrons of HR-TEM were not able to pass through the crystal. Following the reviewer’s suggestion, we have conducted SEM and TEM with EDX line profile of CoO_x/Mo:BiVO₄ and Mo:BiVO₄/Pd. These results further support the facet-selective loadings of CoO_x and Pd.

To further support the selective loading CoO_x on {110} facet, we added the SEM and TEM line profile of CoO_x/Mo:BiVO₄ and revised the manuscript accordingly:

Figure S7. (a) SEM image of CoO_x/Mo:BiVO₄. (b) Line profile along with the yellow arrow of CoO_x/Mo:BiVO₄.

Figure S8. (a) TEM image of CoO_x/Mo:BiVO₄. (b) Line profile along with the white arrow of CoO_x/Mo:BiVO₄.

Line 95: ‘Consistent with the SEM results, energy-dispersive X-ray spectroscopy (EDS) elemental mapping and line profile (Fig. 1e and f) show 4.1-fold stronger Co signal on the

{110} facet compared to that on the {010} facet. Further, SEM (Fig. S7) and TEM (Fig. S8) line profiles of $\text{CoO}_x/\text{Mo:BiVO}_4$ indicate that Co signal on {110} facet is much higher than that on {010} facet. These results demonstrate the selective deposition of Co on the {110} facet of Mo:BiVO_4 .

To further support the selective loading Pd on {010} facet, we added the result SEM and TEM line profile of $\text{Mo:BiVO}_4/\text{Pd}$ and revised the manuscript accordingly:

Figure S13. (a) SEM image of $\text{Mo:BiVO}_4/\text{Pd}$. (b) Line profile along with the yellow arrow of $\text{Mo:BiVO}_4/\text{Pd}$.

Figure S14. (a) TEM image of $\text{Mo:BiVO}_4/\text{Pd}$. (b) Line profile along with the white arrow of $\text{Mo:BiVO}_4/\text{Pd}$.

Line 118: ‘The facet-selective loading of Pd was further demonstrated by the stark contrast between distinctive Pd signal on the {010} facet and negligible Pd signal on the {110} facet from EDS elemental mapping and line profile of Mo:BiVO_4 (Figs. 1e, 1f and S12). SEM (Fig. S13) and TEM (Fig. S14) line profile of $\text{Mo:BiVO}_4/\text{Pd}$ also suggest that Pd signal on {010} facet is much higher than that on {110} facet.’

Finally, we conducted further SEM analysis to support the selective loadings of CoO_x and Pd on {110} and {010} facet, respectively.

Figure S12. SEM images of CoO_x/Mo:BiVO₄/Pd showing selective loadings of CoO_x and Pd distribution on {110} and {010} facet, respectively.

We have revised the manuscript accordingly:

Line 118: ‘The facet-selective loading of Pd was further demonstrated by the stark contrast between distinctive Pd signal on the {010} facet and negligible Pd signal on the {110} facet from EDS elemental mapping and line profile of Mo:BiVO₄ (Figs. 1e, 1f and S12).’

Comment 4.4 *Line 121: The authors mentioned that the enhanced H₂O₂ production selectivity with Pd loading was not perturbed by the presence of CoO_x. However, according to Fig. S15, H₂O₂ production selectivity was perturbed in the presence of CoO_x actually. It is suggested that the authors give corresponding explanations based on subsequent analysis.*

Response – The selectivity of H₂O₂ generation of Mo:BiVO₄/Pd and CoO_x/Mo:BiVO₄/Pd were 89% and 83%, respectively. The decrease on the selectivity after loading CoO_x is 6% which is quite small, so we drew a conclusion “the enhanced H₂O₂ production selectivity with Pd loading was slightly perturbed by the presence of CoO_x” in our original text. We do agree with the reviewer that the H₂O₂ production selectivity was slightly perturbed by CoO_x loading, which is a more accurate description. Therefore, we revised the text to make our discussion more accurate:

Line 128: ‘Moreover, the enhanced H₂O₂ production selectivity with Pd loading was slightly perturbed by the presence of CoO_x, likely attributed to improved H₂O₂ decomposition (Fig. S15). Note that the two-electron H₂ evolution (2H⁺ + 2e⁻ → H₂), another major side reaction likely limiting the selectivity for H₂O₂ generation,¹⁹ was prohibited because the conduction band of Mo:BiVO₄ is too deep to evolve H₂.’

Comment 4.5 *Various comparisons of photocatalytic H₂O₂ generation performance are given elaborately. Since WOR cocatalyst plays an important role in charge separation, the oxygen evolution part of the photocatalytic system should not be ignored. How about the time course of O₂ evolution over CoO_x/BiVO₄:Mo/Pd?*

Response We thank the reviewer for reminding us of this important experiment. We have added the O₂ evolution over CoO_x/Mo:BiVO₄/Pd in Fig. S9 and revised the manuscript accordingly:

Figure S9. (a) Time course of O₂ evolution over Mo:BiVO₄ with or without CoO_x and Pd. Reaction conditions: photocatalyst amount, 0.1 g; reactant solution, 100 mL water with 0.1 g La₂O₃; background atmosphere, water vapor and 50 torr Ar; light source: xemon lamp, 300 W, λ > 420 nm.

Line 203: ‘In the meantime, when CoO_x and Pd were randomly deposited on Mo:BiVO₄ (denoted as Mo:BiVO₄-CoO_x-Pd with a SEM image in Fig. S22), it was only 32% as active as CoO_x/Mo:BiVO₄/Pd, though these two catalysts had similar surface kinetics and selectivity. Further, the photocatalytic O₂ evolution activity of CoO_x/Mo:BiVO₄/Pd was 3.2-fold higher than that of CoO_x/Mo:BiVO₄ (Fig. S9a). These results suggest that selective coloaded of CoO_x and Pd not only improved surface kinetics for O₂ evolution and selectivity for H₂O₂ production as introduced above, but also tuned other critical processes like charge separation.³¹’

Comment 4.6 *Line 143-line 146: More evidence should be provided for high resistance to •OH-mediated oxidation of CoO_x/Mo:BiVO₄/Pd, such as XPS spectra of BiVO₄ before and after action.*

Response

We thank the reviewer for the opportunity to improve our discussions. Following the reviewer’s suggestion, we compared the XPS spectrum before and after •OH-mediated oxidation. The result shows no difference on Bi, V and O XPS spectra, confirming the high stability of CoO_x/Mo:BiVO₄/Pd. We have added these results and related discussions in the revised manuscript and SI:

Figure S17. The Bi, V and O XPS spectra of CoO_x/Mo:BiVO₄/Pd before and after •OH aging

for 24h.

Line 152: ‘Most importantly, in stark contrast to the high •OH-susceptibility of organic photocatalyst, after 24-h incubation under •OH-rich conditions, CoO_x/Mo:BiVO₄/Pd exhibits nominal change in H₂O₂ productions (Fig. S1b) and chemical compositions (Fig. S17), demonstrating its high resistance to •OH-mediated oxidation.’

Comment 4.7 Line 163: The authors believed that CoPi solely facilitated the surface kinetics for O₂ evolution and hardly affected the energetics for charge separation. If CoPi had no adverse effects, why the photocatalytic H₂O₂ generation rate of CoO_x/Mo:BiVO₄/Pd decreased gradually?

Response Firstly, both CoO_x and CoPi facilitate surface kinetics. However, while CoO_x is confirmed to improve charge separation, CoPi is not able to improve charge separation (*J. Am. Chem. Soc.*, 2012, 134, 16693–16700). Since the charge separation efficiency of CoPi/Mo:BiVO₄/Pd is lower than CoO_x/Mo:BiVO₄/Pd, the photocatalytic H₂O₂ generation rate of CoO_x/Mo:BiVO₄/Pd decreased because CoO_x is gradually converted to CoPi.

Secondly, we would like to note the difference between the stability of photocatalyst and H₂O₂ production. The photocatalyst CoO_x/Mo:BiVO₄/Pd itself is actually very stable and no deactivation was observed in pure water over repetitive use:

Figure S18b. Time courses of photocatalytic H₂O₂ generation over CoO_x/Mo:BiVO₄/Pd in pure water.

We further tested the performance of CoO_x/Mo:BiVO₄/Pd in seawater which is a desirable solution condition for artificial photosynthesis (*J. Phys. Chem. Lett.*, 2010, 1, 2655–2661). No deactivation was observed over five-round repetitive use:

Figure S21. Time courses of photocatalytic H_2O_2 generation over $\text{CoO}_x/\text{Mo}:\text{BiVO}_4/\text{Pd}$ in seawater. The seawater is prepared by dissolving the sea salt consisted with NaCl (77.6 wt%), KCl (2.1 wt%), MgCl_2 (6.6 wt%), CaCl_2 (3.3 wt%), MgSO_4 (9.6 wt%), NaHCO_3 (0.6 wt%), NaBr (0.16 wt%), Na_2SiO_3 (0.007 wt%), NaSi_4O_9 (0.006 wt%), H_3PO_4 (0.006 wt%), H_3BO_3 (0.2 wt%), LiNO_3 (0.004 wt%), Al_2Cl_6 (0.04 wt%) in the DI water.

To provide further evidence for photocatalyst stability, we compared the XPS spectrum before and after $\bullet\text{OH}$ -mediated oxidation. The result shows no difference on Bi, V and O XPS spectra, confirming the high stability of $\text{CoO}_x/\text{Mo}:\text{BiVO}_4/\text{Pd}$. We have added these results and related discussions in the revised manuscript and SI:

Figure S17. The Bi, V and O XPS spectra of $\text{CoO}_x/\text{Mo}:\text{BiVO}_4/\text{Pd}$ before and after $\bullet\text{OH}$ aging for 24h.

Line 152: ‘Most importantly, in stark contrast to the high $\bullet\text{OH}$ -susceptibility of organic photocatalyst, after 24-h incubation under $\bullet\text{OH}$ -rich conditions, $\text{CoO}_x/\text{Mo}:\text{BiVO}_4/\text{Pd}$ exhibits nominal change in H_2O_2 productions (Fig. S1b) and chemical compositions (Fig. S17), demonstrating its high resistance to $\bullet\text{OH}$ -mediated oxidation.’

Moreover, as shown in Fig. S18b and also observed in many other H_2O_2 photosynthesis studies, the cumulative H_2O_2 production over time is not linear, which is caused by H_2O_2 decomposition rather than the deactivation of photocatalysts. Therefore, PO_4^{3-} was applied to inhibit the decomposition of H_2O_2 , and to improve the accuracy of activity tests. The addition of PO_4^{3-} is quite common in H_2O_2 production performance tests, for instance:

Shiraishi, Y., Ueda, Y., Soramoto, A., Hinokuma, S. & Hirai, T. Photocatalytic hydrogen peroxide splitting on metal-free powders assisted by phosphoric acid as a stabilizer. *Nat.*

Commun. **11** (2020).

Greenspan, F. P. Method of improving stability of concentrated hydrogen peroxide in contact with stainless and aluminum alloys. US patent US2782100A (1957).

Shimokawa, S., Namikawa, K. & Murakami, S. Stabilized refined aqueous hydrogen peroxide solution. Jpn. Kokai Tokkyo Koho JP07-081906A (1995).

For large-scale applications of a particulate photocatalyst (also a potential research direction for applying our $\text{CoO}_x/\text{Mo}:\text{BiVO}_4/\text{Pd}$ photocatalyst), the photocatalyst is typically immobilized on a support using drop-casting (*Energy Technol-Ger*, 2015, **3**, 1014–1017) or screen printing (*Nat. Mater.*, 2016, **15**, 611–615) technologies. Then, the reaction will be conducted with flow liquid. In those setups, the photogenerated H_2O_2 will be immediately removed from catalyst surface to prevent H_2O_2 decomposition. Naturally, the addition of PO_4^{3-} and resultant transformation of CoO_x to CoPi will be avoided. For instance, in our previous studies, we fabricated a 1×1 m $\text{Al}:\text{SrTiO}_3$ panel by drop-casting as a prototype for large-scale solar hydrogen generation (see movie: <https://ars.els-cdn.com/content/image/1-s2.0-S2542435117302246-mmc3.mp4>; *Joule*, 2018, **2**, 509–520). By immobilizing the photocatalyst, packing of photocatalyst was prohibited and photocatalyst showed stable activity for at least 7h under the intense illumination as an accelerated deactivation test (*Joule*, 2018, **2**, 509–520). In our recent work, the feasibility of applying immobilized photocatalyst with high solar-to-chemical energy conversion efficiency was demonstrated on a 100-m^2 scale (*Nature*, 2021, 598, 304–307).

Referring to our previous large-scale photolysis device setup, we plan to fabricate the $\text{CoO}_x/\text{Mo}:\text{BiVO}_4/\text{Pd}$ panel where SiO_2 nanoparticles (diameter ca. 20 nm) will be used as an inorganic binder to firmly fix a $\text{CoO}_x/\text{Mo}:\text{BiVO}_4/\text{Pd}$ layer onto the substrate (e.g., glass, *Joule*, 2018, **2**, 509–520). The device validation and performance test are being conducted in ongoing works. As a first step, we designed a flow cell photolysis system, where photogenerated H_2O_2 is immediately separated from $\text{CoO}_x/\text{Mo}:\text{BiVO}_4/\text{Pd}$ particles to avoid H_2O_2 decomposition, and therefore no PO_4^{3-} addition is required.

Figure R1. Photograph and schematic illustration of the flow cell device for photocatalytic H_2O_2 productions.

We have added related discussions in the revised manuscript to clarify the stability of CoO_x/Mo:BiVO₄/Pd:

Line 185: ‘As expected, CoO_x/Mo:BiVO₄/Pd was highly stable over five cycles of reaction. To further demonstrate the stability of CoO_x/Mo:BiVO₄/Pd, we test the performance of CoO_x/Mo:BiVO₄/Pd in seawater which is a desirable solution condition for artificial photosynthesis.²⁸ No deactivation was observed over five-round repetitive use (Fig. S21). Yet the cumulative production of H₂O₂ was lower than that in phosphate solution owing to H₂O₂ decomposition, consistent with the previous report.²⁹

Line 188: ‘Yet the cumulative production of H₂O₂ was lower than that in phosphate solution owing to H₂O₂ decomposition, consistent with the previous report.²⁹ In following studies, the cumulative production of H₂O₂ in pure water will be improved with rapid H₂O₂ diffusion by a large-scale photosynthesis setup where the CoO_x/Mo:BiVO₄/Pd photocatalyst will be immobilized on a support using drop-casting or screen printing technologies and integrated in a flow cell photolysis system.^{19,30}

Line 463: ‘30. Schroder, M. *et al.* Hydrogen evolution reaction in a large-scale reactor using a carbon nitride photocatalyst under natural sunlight irradiation. *Energy Technol-Ger* **3**, 1014-1017 (2015).’

Reviewer 5

Comment 5.1 *In this article, titled, “Overall photosynthesis of H₂O₂ by an inorganic semiconductor”, the authors develop a robust all-inorganic photocatalyst system (CoO_x on {110}/ Mo:BiVO₄/ Pd on {010}) for the photocatalytic synthesis of peroxide from water (STC of ~0.29%). In this article, the authors synthesise nanoparticulate Mo-doped BiVO₄ with distinct {110} and {010} facets. The co-catalysts CoO_x and Pd were selective deposited on the {110} and {010} facets, respectively. The physical properties of the photocatalyst, and its parent materials, were well characterised using XRD, SEM with EDS, TEM & SAED, XPS etc. The photocatalytic activity of the photocatalyst, and its parent materials, were also well characterised, with the activities expressed as apparent quantum yields (AOY) at a fixed wavelength, full spectrum quantum yield and solar-to-chemical conversion efficiency for the selective formation of H₂O₂ from splitting water. Also, the photophysical behaviour of the photocatalyst, and its parent materials, were explored using ultra-fast and slow timescale TAS, which provided evidence for the charge transfer of holes and electrons to the CoO_x and Pd co-catalysts, respectively, and its benefits with respect to increases in charge carrier lifetimes. Overall, the work is of a very high standard, and I recommend its publication in your journal (minor corrections). Below there are some typos from the Abstract and Introduction sections and some specific queries that I would like addressed before the final submission.*

Response We thank the reviewer for the time and effort to review our manuscript as well as the encouraging comments. We are humbled by the reviewer’s praises.

Comment 5.2 *“on proper facet” should be “on distinct facets”*

Response We have revised the manuscript accordingly:

Line 20: ‘Time-resolved spectroscopic investigations of photocarriers suggest that depositing select cocatalysts on **distinct** facet tailored the interfacial energetics between {110} and {010} facets and enhanced charge separation in **Mo:BiVO₄**, therefore overcoming a key challenge in developing efficient inorganic photocatalysts.’.

Comment 5.3 *“With regard of reaction” should be “With regard to reaction”*

Response We have revised the manuscript accordingly:

Line 33: ‘With regard **to** reaction processes, PC systems are advantageous **for the** mass transport of reagents and products, which greatly reduces **the** concentration overpotential and pH gradient during reactions.’.

Comment 5.4 *“are advantageous on mass” should be “are advantageous for the mass”*

Response We have revised the manuscript accordingly:

Line 33: ‘With regard to reaction processes, PC systems are advantageous for the mass transport of reagents and products, which greatly reduces the concentration overpotential and pH gradient during reactions.’

Comment 5.5 “greatly reduces concentration” should be “greatly reduces the concentration”.

Response We have revised the manuscript accordingly:

Line 33: ‘With regard to reaction processes, PC systems are advantageous for the mass transport of reagents and products, which greatly reduces the concentration overpotential and pH gradient during reactions.’

Comment 5.6 “reported efficiency remain unsatisfying” should be “reported efficiency remains unsatisfying”.

Response We have revised the manuscript accordingly:

Line 52. ‘BiVO₄ is a type of photocatalysts with a suitable band structure and relatively narrow bandgap (2.4 eV) for H₂O₂ photosynthesis, yet the reported efficiency remains unsatisfying (< 12 μM/h, see Table S1) due to severe charge recombination, even in the presence of sacrificial agents.’

Comment 5.7 “due to severe charge recombination even” should be “due to severe charge recombination, even”.

Response We have revised the manuscript accordingly:

Line 52. ‘BiVO₄ is a type of photocatalysts with a suitable band structure and relatively narrow bandgap (2.4 eV) for H₂O₂ photosynthesis, yet the reported efficiency remains unsatisfying (< 12 μM/h, see Table S1) due to severe charge recombination, even in the presence of sacrificial agents.’

Comment 5.8 “leveraging the record” should be “surpassing the record”.

Response We have revised the manuscript accordingly:

Line 62. ‘Without using any sacrificial reagent, the reasonably designed CoO_x/Mo:BiVO₄/Pd produced H₂O₂ at a rate of 1425 μM/h, an apparent quantum yield (AQY) of 1.2% over the full spectrum of sunlight, and a STH of 0.29%, surpassing the record of inorganic photocatalysts by one order of magnitude (Table S1).’

Comment 5.8 I have only provided typos for the Abstract and Introduction section. I recommend that the entire article is proof read once more before its final submission.

Response We have revised the grammatical errors that were pointed out. We further revised the whole manuscript accordingly. The revision involving grammatical errors were

highlighted in purple.

Comment 5.9 *I don't know what the convention is for Nature Communications, but from what I have seen, it is more usual to write, "Mo:BiVO₄" than "BiVO₄:Mo" (see Google Ngram Viewer statistics on this). I therefore suggest you use Mo:BiVO₄ throughout.*

Response We thank the reviewer's suggestion. We have rewritten all the 'BiVO₄:Mo' as 'Mo:BiVO₄' in the whole manuscript.

Comment 5.10 *In your XPS analysis section, you write "The Pd 3d_{5/2} peak can be deconvoluted to a main Pd⁰ peak at 335.1 eV and a minor Pd²⁺ peak at 337.0 eV, attributing to the metallic Pd from photoreduction and PdO₂ from partial oxidation of Pd in air, respectively." Do you mean PdO, as the oxidation state of Pd is 2+?*

Response We agree with the reviewer that Pd oxidation state should be 2+. We have revised the manuscript accordingly:

Line 114: 'The Pd 3d_{5/2} peak can be deconvoluted to a main Pd⁰ peak at 335.1 eV and a minor Pd²⁺ peak at 337.0 eV, attributing to the metallic Pd from photoreduction and PdO from partial oxidation of Pd in air, respectively.'

Comment 5.11 *You achieve an AQY at 420 nm of 5.8%, which is the highest reported for inorganic semiconductors to date. I recommend that this finding be highlighted in the abstract.*

Response We thank the reviewer for the suggestion on highlighting our findings. We have added the information in the revised abstract accordingly:

Line 15: 'Here, we report an inorganic Mo-doped faceted BiVO₄ (Mo:BiVO₄) system that is resistant to radical oxidation and exhibits a record overall H₂O₂ photosynthesis efficiency among inorganic photocatalysts, with an apparent quantum yield of 1.2% and a solar-to-chemical conversion efficiency of 0.29% at full spectrum, as well as an apparent quantum yield of 5.8% at 420 nm. The surface reaction kinetics and selectivity of Mo:BiVO₄ were tuned by precisely loading CoO_x and Pd on {110} and {010} facets, respectively.'

Comment 5.12 *In Figure 2a, the data for Mo:BiVO₄ overlaps with that of CoO_x/Mo:BiVO₄, so it is hard to see. I'm not sure what is the best way to resolve this. Perhaps make one of the data series 50% transparent so that the other can be seen, or state in the figure caption that data from these two series overlap.*

Response We thank the reviewer for the opportunity to improve the clarity. We have revised the caption accordingly

Figure 2a. Time courses of photocatalytic H₂O₂ generation over CoO_x/Mo:BiVO₄/Pd, Mo:BiVO₄-CoO_x-Pd, CoO_x/Mo:BiVO₄, Mo:BiVO₄/Pd, and Mo:BiVO₄. Reaction conditions: photocatalyst amount, 2 mg; reactant solution, 12 ml PBS aqueous solution (pH=7.4) saturated with O₂; light source, xenon lamp solar simulator, 100 mW/cm², AM 1.5G. We note the data series for CoO_x/Mo:BiVO₄ (7.5 μM at the reaction time of 60 min) overlap with those of Mo:BiVO₄ (4.1 μM at the reaction time of 60 min).

Comment 5.13 You write, “The cumulative production of H₂O₂ in pure water will be improved by a large scale photosynthesis setup with rapid H₂O₂ diffusion in a following study.” I’m not sure what you mean by rapid diffusion here. Please make this more clear to the reader how the rapid diffusion of H₂O₂ will prevent its decomposition in future work.

Response We thank the reviewer for bringing up this important point and opportunity to improve the clarity. In a typical batch setup, the cumulative production of H₂O₂ slows down over time due to simultaneous H₂O₂ decomposition on the surface of photocatalyst. Such H₂O₂ decomposition can be solved by separation of H₂O₂ and photocatalyst after H₂O₂ production. For instance, for large-scale applications of a particulate photocatalyst (also a potential research direction for applying our CoO_x/Mo:BiVO₄/Pd photocatalyst), the photocatalyst is typically immobilized on a support using drop-casting (*Energy Technol-Ger*, 2015, **3**, 1014–1017) or screen printing (*Nat. Mater.*, 2016, **15**, 611–615) technologies. Then, the reaction will be conducted with flow liquid. In those setups, the photogenerated H₂O₂ will be immediately removed from catalyst surface to prevent H₂O₂ decomposition. For instance, in our previous studies, we fabricated a 1 × 1 m Al:SrTiO₃ panel by drop-casting as a prototype for large-scale solar hydrogen generation (see movie: <https://ars.els-cdn.com/content/image/1-s2.0-S2542435117302246-mmc3.mp4>; *Joule*, 2018, **2**, 509–520). By immobilizing the photocatalyst, packing of photocatalyst was prohibited and photocatalyst showed stable activity for at least 7h under the intense illumination as an accelerated deactivation test (*Joule*, 2018, **2**, 509–520). In our recent work, the feasibility of applying immobilized photocatalyst with high solar-to-chemical energy conversion efficiency was demonstrated on a 100-m² scale (*Nature*, 2021, 598, 304–307).

Referring to our previous large-scale photolysis device setup, we plan to fabricate the

CoO_x/Mo:BiVO₄/Pd panel where SiO₂ nanoparticles (diameter ca. 20 nm) will be used as an inorganic binder to firmly fix a CoO_x/Mo:BiVO₄/Pd layer onto the substrate (e.g., glass, *Joule*, 2018, 2, 509-520). The device validation and performance test are being conducted in ongoing works. As a first step, we designed a flow cell photolysis system, where photogenerated H₂O₂ is immediately separated from CoO_x/Mo:BiVO₄/Pd particles to avoid H₂O₂ decomposition, and therefore no PO₄³⁻ addition is required.

Figure R1. Photograph and schematic illustration of the flow cell device for photocatalytic H₂O₂ productions.

We have added related discussions in the revised manuscript to clarify the flow cell photolysis setup to improve H₂O₂ stability:

Line 188: ‘Yet the cumulative production of H₂O₂ was lower than that in phosphate solution owing to H₂O₂ decomposition, consistent with the previous report.²⁹ In following studies, the cumulative production of H₂O₂ in pure water will be improved with rapid H₂O₂ diffusion by a large-scale photosynthesis setup where the CoO_x/Mo:BiVO₄/Pd photocatalyst will be immobilized on a support using drop-casting or screen printing technologies and integrated in a flow cell photolysis system.’^{19,30}

Comment 5.14 *You write, “We note that the enhancement on H₂O₂ production by coloading Pd and CoO_x is even higher than the multiplication of individual enhancements”. Perhaps use the term ‘synergistic’.*

Response We thank the reviewer for the suggestion. We have revised the manuscript accordingly:

Line 200: ‘We note that the enhancement of H₂O₂ production by the synergistic effect of coloading Pd and CoO_x is even higher than the multiplication of the enhancements by loading Pd and CoO_x individually, i.e., 347.6-fold for coloading Pd and CoO_x, 53.7-fold for solely loading Pd, and 1.8-fold for solely loading CoO_x.’

Comment 5.15 *In your experimental section, you do not explain if your samples were investigated in air or in solution for your transient absorption spectroscopy studies. Please state the medium in which you samples were measured in the experimental, and also, the figure caption (Figure 4).*

Response We thank the reviewer for this suggestion. We have added the details of TAS

measurements in the manuscript:

Line 385: ‘The decay curves were obtained at 10 ps intervals and accumulated signals were averaged over 1000 – 4000 scans for one point. TAS measurements were performed in vacuum (base pressure $\sim 10^{-5}$ Torr). For sample preparation, each Mo:BiVO₄ and (Pd, CoO_x)-loaded Mo:BiVO₄ powders was prepared by dispersing the powder on isopropanol and then drop-casted on a circular CaF₂ substrate and subsequently dried naturally in air to obtain a powder film with a density of ~ 1.25 mg cm⁻².’

Comment 5.16 In Figure 4c you show band energies and band bending in your photocatalysts loading with co-catalysts. You also assume that the energy of the {110} facet is more negative than that of the {010} facet. How were these energies determined? (facets, Fermi energies of the co-catalysts, band-bending etc.). Please provide some details of this calculation in your experimental section.

Response We thank the reviewer for raising this important discussion point. The facet dependent energetics was first confirmed on anatase TiO₂ and such phenomenon was caused by the variation in surface termination atoms (*Nature* 2008, 453, 638–641). While the origin of the band bending between {110} and {010} facets on BiVO₄ has not been investigated, we believe that it will be similar with that for faceted TiO₂.

The band bending exists on both the surface and the bulk of BiVO₄. The band bending extends from surface to bulk, constructing a space-charge region. Such band bending phenomenon has been nicely imaged by a recent work from Can Li’s group using spatially resolved surface photovoltage spectroscopy (see below) (*Nat. Commun.*, 2013, 4, DOI: 10.1038/ncomms2401; *Angew. Chem. Int. Edit.*, 2015, 54, 9111–9114; *Nano Lett.*, 2017, 17, 6735–6741). We would like to emphasize that the spatial charge separation of a single crystal is attributed to the band bending of different facets, and loading a cocatalyst can tune the surface energetics for enhancing charge separation.

Figure R2. Schematic of the built-in electric field with relative strength in the SCR of different facets (*Angew. Chem. Int. Edit.*, 2015, **54**, 9111-9114).

To further illustrate the banding bending between different facets, we have added a new scheme in the revised SI and revised the manuscript:

Figure S25. Schematic band diagrams across the border between the {011} and {010} facets of a bare single Mo:BiVO₄ photocatalyst particle and facets of a single Mo:BiVO₄ photocatalyst particle with CoO_x cocatalyst selectively deposited on {010} facet (green line) and Pd cocatalyst selectively deposited on {010} facet (orange line), respectively. The facet dependent energetics on a bare single Mo:BiVO₄ photocatalyst particle was caused by the variation in surface termination atoms.¹⁶

Line 280: ‘Such positive effects are supposed to be achieved by tuning the energetics between cocatalysts and respective Mo:BiVO₄ facet (Figs. 4c and S25).’

Comment 5.17 In your experimental section on “Photocatalytic activity tests”, please provide details, if any, on the calibration of your H₂O₂ measurement set-up using known concentrations of H₂O₂ from a stock solution. Please provide this curve, if any, in your supporting information (apologies if this information is already included, and I have missed it).

Response We thank the reviewer for the opportunity to improve the clarity. Following the reviewer’s suggestion, we have added the more H₂O₂ measurement details and calibration curve in the revised SI and revised manuscript accordingly:

Figure S30. Calibration curve for quantifying photogenerated H₂O₂. The H₂O₂ concentration

was calculated following the equation: $\text{Counts} = 365.2[\text{H}_2\text{O}_2] + 120$. For example, HPLC analysis of 60-min H_2O_2 production by $\text{CoO}_x/\text{Mo}:\text{BiVO}_4/\text{Pd}$ gave a signal of 6620, corresponding to a H_2O_2 concentration of 17.8 μM . Because we diluted the H_2O_2 solution by 80 times before analysis, the photogenerated H_2O_2 was 1425 μM .

Line 333: 'Resorufin in the mixture solution was quantified using an Agilent high-performance liquid chromatography coupled to a photo-diode array detector (detection at 560 nm); 50 μL of each sample was injected. The calibration in Fig. S30 is used to quantitatively analyze the H_2O_2 concentration. Separation was carried out in a C18 column at 20 $^\circ\text{C}$ with an isocratic mobile phase of 55% sodium citrate buffer (with 10% methanol (v/v), pH 7.4) and 45% methanol (v/v) at a flow rate of 0.5 mL min^{-1} .'

Line 326: 'For the wavelength-dependent AQY analysis, the photolysis was performed using LED light irradiation (model slight; Perfect Light, Inc.). At designated time points, 50 μL suspension was taken for analysis of H_2O_2 productions and diluted with phosphate buffer (pH=7.4) to a H_2O_2 concentration (2-20 μM) that is most suitable for accurate H_2O_2 quantification, followed by centrifugation. Then 50 μL supernatant was taken and mixed with 50 μL solutions containing phosphate buffer (50 mM, pH=7.4), ampliflu red (100 μM) and horseradish peroxidase (0.05 U/mL).'

Additional Correction

We have corrected a typo in Figure S9a. The unit of gas evolution should be μmol instead of mmol . This change does affect the conclusion of this work.

Figure S9. (a) Time course of O_2 evolution over $\text{Mo}:\text{BiVO}_4$ with or without CoO_x and Pd. Reaction conditions: photocatalyst amount, 0.1 g; reactant solution, 100 mL water with 0.1 g La_2O_3 ; background atmosphere, water vapor and 50 torr Ar; light source: Xenon lamp, 300 W, $\lambda > 420$ nm.

We have acknowledged help in additional catalyst characterizations.

Line 392: 'We are grateful to Sudan Shen (State Key Laboratory of Chemical Engineering at Zhejiang University) for help in TEM measurements.'

REVIEWERS' COMMENTS

Reviewer #1 (Remarks to the Author):

The authors have addressed the concerns adequately. The clarity of the work has been improved considerably. The reviewer can recommend acceptance of this revised manuscript.

Reviewer #2 (Remarks to the Author):

Overall photosynthesis of H₂O₂ by an inorganic semiconductor

T. Liu et al.

The authors appropriately responded to all the comments and suggestions for the original paper. Then, I can recommend publication of this paper in Nature Communications.

* The sentence on Lines 270-272 is grammatically incorrect.

Reviewer #3 (Remarks to the Author):

This manuscript reports on a very nice piece of works dealing with the use of CoO_x/BiVO₄:Mo/Pd heteromicrostructures for the efficient production of H₂O₂ by photocatalysis under solar light. As it was mentioned in the initial reviewer's report, it is a very robust study reporting excellent photocatalytic performances and implementing the most recent characterization techniques to rationalize the photocatalytic properties of the heterostructures prepared. In addition, the authors have carefully answered the points raised by the reviewer in his previous report. As a consequence, the reviewer recommends the publication of this revised manuscript in the present state.

Reviewer #4 (Remarks to the Author):

Congratulateion. I don't have additional comments.

Reviewer #5 (Remarks to the Author):

All queries I have raised (Reviewer 5) have been adequately addressed, and I recommend that this paper for publication.

Response to Comments

Overall photosynthesis of H₂O₂ by an inorganic semiconductor

Tian Liu, Zhenhua Pan, Junie Jhon M. Vequizo, Kosaku Kato, Binbin Wu, Akira Yamakata, Kenji Katayama, Baoliang Chen, Chiheng Chu, Kazunari Domen

Reviewer 1

Comment 1.1 *The authors have addressed the concerns adequately. The clarity of the work has been improved considerably. The reviewer can recommend acceptance of this revised manuscript.*

Response We thank the reviewer very much for the thorough review, which help to improve the clarity and quality of our manuscript.

Reviewer 2

Comment 2.1 *The authors appropriately responded to all the comments and suggestions for the original paper. Then, I can recommend publication of this paper in Nature Communications.*

Response We thank the reviewer very much for the time and effort to improve the quality of our manuscript as well as the encouraging comments.

Comment 2.2 *The sentence on Lines 270-272 is grammatically incorrect.*

Response We have revised the manuscript accordingly:

Line 271: 'This result suggests that the deeply trapped electrons transferred to Pd and became available for subsequent surface reactions in Mo:BiVO₄/Pd and CoO_x/Mo:BiVO₄/Pd?.'

Reviewer 3

Comment 3.1 *This manuscript reports on a very nice piece of works dealing with the use of CoO_x/Mo:BiVO₄/Pd heteromicrostructures for the efficient production of H₂O₂ by photocatalysis under solar light. As it was mentioned in the initial reviewer's report, it is a very robust study reporting excellent photoalytic performances and implementing the most recent characterization techniques to rationalize the photocatalytic properties of the heterostructures prepared. In addition, the authors have carefully answered the points raised by the reviewer in his previous report. As a consequence, the reviewer recommends the publication of this revised manuscript in the present state.*

Response We thank the reviewer very much for the time and effort to improve the quality of our manuscript as well as the encouraging comments.

Reviewer 4

Comment 4.1 Congratulation. I don't have additional comments.

Response We thank the reviewer very much for the time and effort to improve the quality of our manuscript as well as the encouraging comments.

Reviewer 5

Comment 5.1 All queries I have raised have been adequately addressed, and I recommend that this paper for publication.

Response We thank the reviewer very much for the time and effort to improve the quality of our manuscript as well as the encouraging comments.

Additional Correction

We have corrected the caption in Figure S23. The 'Figure S16' should be 'Figure S24'.

Figure S23. TA spectra of Mo:BiVO₄ irradiated with laser pulses (duration: 6 ns, fluence: 3 mJ pulse⁻¹) in vacuum. The strong absorption at 20000 – 17000 cm⁻¹ is attributed to trapped holes as reported previously on Mo:BiVO₄ photocatalysts.¹²⁻¹⁴ The broad absorption from 170000 – 5000 cm⁻¹ is assigned to deeply trapped electrons since the TA intensity at this region decreased when Pd as an electron sink was loaded on Mo:BiVO₄ (Figure S24). The absorption < 5000 cm⁻¹ was reported to be free/shallowly trapped electrons.¹⁵